# Characterization of a *Cutibacterium acnes* Camp Factor 1-Related Peptide as a New TLR-2 Modulator in In Vitro and Ex Vivo Models of Inflammation

**DOI:** 10.3390/ijms23095065

**Published:** 2022-05-03

**Authors:** Constance Mayslich, Philippe Alain Grange, Mathieu Castela, Anne Geneviève Marcelin, Vincent Calvez, Nicolas Dupin

**Affiliations:** 1Département DRC, Développement, Reproduction et Cancer, Institut Cochin, INSERM U1016-CNRS UMR8104, Université Paris Cité, 75014 Paris, France; constance.mayslich@inserm.fr (C.M.); philippe.grange@aphp.fr (P.A.G.); mathieu.castela@scarcell.com (M.C.); 2Service de Dermatologie-Vénéréologie et CeGIDD, Groupe Hospitalier APHP.centre, CNR IST Bactériennes—Laboratoire Associé Syphilis, 75014 Paris, France; 3Hôpital Cochin, U1016, Equipe Biologie Cutanée—CNR IST bactériennes—Syphilis 24, rue du faubourg Saint-Jacques, 75014 Paris, France; 4National Reference Centre for Herpesviruses, Virology Department, Team 3 THERAVIR, and AP-HP, Pitié-Salpêtrière—Charles Foix University Hospital, Institut Pierre Louis d’Epidémiologie et de Santé Publique (iPLESP), INSERM, Sorbonne Université, 75013 Paris, France; anne-genevieve.marcelin@aphp.fr (A.G.M.); vincent.calvez@aphp.fr (V.C.)

**Keywords:** *C. acnes*, virulence factors, CAMP factors, binding, TLR-2, inflammation

## Abstract

*Cutibacterium acnes* (*C. acnes*) has been implicated in inflammatory acne where highly mutated Christie–Atkins–Munch–Petersen factor (CAMP)1 displays strong toll like receptor (TLR)-2 binding activity. Using specific antibodies, we showed that CAMP1 production was independent of *C. acnes* phylotype and involved in the induction of inflammation. We confirmed that TLR-2 bound both mutated and non-mutated recombinant CAMP1, and peptide array analysis showed that seven peptides (A14, A15, B1, B2, B3, C1 and C3) were involved in TLR-2 binding, located on the same side of the three-dimensional structure of CAMP1. Both mutated and non-mutated recombinant CAMP1 proteins induced the production of C-X-C motif chemokine ligand interleukin (CXCL)8/(IL)-8 in vitro in keratinocytes and that of granulocyte macrophage-colony stimulating factor (GM-CSF), tumor necrosis factor (TNF)-α, IL-1β and IL-10 in ex vivo human skin explants. Only A14, B1 and B2 inhibited the production of CXCL8/IL-8 by keratinocytes and that of (GM-CSF), TNF-α, IL-1β and IL-10 in human skin explants stimulated with rCAMP1 and *C. acnes*. Following pretreatment with B2, RNA sequencing on skin explants identified the 10 genes displaying the strongest differential expression as *IL6*, *TNF*, *CXCL1*, *CXCL2*, *CXCL3*, *CXCL8*, *IL-1β*, chemokine ligand (*CCL*)*2*, *CCL4* and colony stimulating factor (*CSF*)*2*. We, thus, identified a new CAMP1-derived peptide as a TLR-2 modulator likely to be a good candidate for clinical evaluation.

## 1. Introduction

The Gram-positive bacterium *Cutibacterium acnes* (*C. acnes*), previously known as *Propionibacterium acnes*, is a commensal, anaerobic, aerotolerant, lipophilic bacterium found mostly in sebum-rich areas within the pilosebaceous unit (PSU) in normal human skin [1,2]. *C. acnes* strains are currently classified into eight phylotypes (IA-1, IA-2, IB-1, IB-2, IB-3, IC, II, III), with the IA-1 and IA-2 clades preferentially found in acne lesions, whereas phylotypes II and III are found in deeper infections [3,4,5,6,7].

*C. acnes* has been implicated in inflammatory acne, but its pathophysiological role remains a matter of debate. Until recently, it was thought that the obstruction of the pilosebaceous unit was due to an increase in sebum production associated with hyperkeratinization, favoring hypoxic conditions and resulting in the proliferation of *C. acnes* strains; however, the *C. acnes* load is similar in PSUs from healthy and acne-affected skin [8]. Moreover, *C. acnes* appears to be the major bacterial genus present on skin. It plays a crucial role in the maintenance of skin health, whereas dysbiosis in the cutaneous microbiota can lead to the selection of specific *C. acnes* lineages, such as the IA1/IA2 phylotypes found predominantly in acne lesions. This phenomenon could be amplified by hyperseborrhea and the change in sebum composition observed in acne, with an increase in proinflammatory lipid content [9,10,11].

However, comparisons of the genomes of *C. acnes* phylotype IA1 isolates from healthy individuals and acne patients revealed no differences [12]. We can, therefore, hypothesize that the expression of specific components by *C. acnes* strains may influence their capacity to colonize the host and stimulate the host’s innate immune system [13]. *C. acnes* is currently considered to be an opportunistic pathogen of low pathogenicity, as it is also isolated in deeper infections, such as endocarditis and postsurgical infections. *C. acnes* strains can induce inflammatory responses via the toll like receptor (TLR)-2 and TLR-4 pathways, by activating the nuclear factor kappa B (NF-κB) and mitogen-activated protein kinase (MAPK) cascades, thereby inducing the production of proinflammatory molecules, such as interleukin (IL)-1α/β, IL-6, C-X-C motif chemokine ligand interleukin (CXCL)8/(IL)-8, IL-12, interferon (IFN), granulocyte macrophage-colony stimulating factor (GM-CSF), tumor necrosis factor (TNF)-α and human beat defensin (hBD)-2 in keratinocytes, sebocytes and monocytes in vitro, but also in acne lesions ex vivo [14,15,16,17,18,19,20].

*C. acnes* strains can produce virulence factors (biofilm, surface proteins), triggering the immune response in the host or enhancing the ability of *C. acnes* to adapt to its environment [8,21]. Several genes encoding putative virulence factors have been identified in the *C. acnes* genome [22], and transcriptomic analysis revealed strong expression of potential virulence factors, such as the dermatan-sulfate adhesins (DsA1 and DsA2), the Christie–Atkins–Munch–Petersen (CAMP) hemolytic factors, the polyunsaturated fatty acid isomerase, the HtaA iron acquisition protein and GehA lipase and heat-shock proteins, such as HSP20, DnaK, DnaJ, GrpE and GroEL [23]. We previously characterized the *C. acnes* hemolytic factor CAMP1, and showed that it was specifically recognized by TLR-2, with a high degree of polymorphism of the CAMP1 protein sequence associated with strong CAMP1-TLR-2 binding in *C. acnes* strains producing large amounts of CXCL8/IL-8 in vitro [24].

In this study, we produced polyclonal antibodies directed against the CAMP1 protein, which we used to analyze the expression levels of this protein in *C. acnes* surface protein extracts. We also cloned and produced non-mutated and mutated CAMP1 proteins and assessed their ability to induce the production of proinflammatory molecules in vitro in keratinocytes and ex vivo in human skin explants. Finally, we identified the CAMP1 peptide sequences involved in TLR-2 binding and CAMP1-related peptides capable of decreasing TLR-2 binding and the production of proinflammatory molecules in vitro and ex vivo.

## 2. Results

### 2.1. Levels of CAMP1 Protein in C. acnes Strains

We previously showed that the CAMP1 protein was recognized only weakly, if at all, by TLR-2 in *C. acnes* phylotype IA1 [24]. We investigated the CAMP1 protein production in *C. acnes* strains, by generating a polyclonal antibody against two 10-mer peptides derived from the CAMP1 protein sequence (Appendix A). Total surface proteins were extracted from 23 *C. acnes* strains and subjected to Western blotting with an anti-CAMP1 antibody, which recognized a protein band (Figure 1). CAMP1 was detected with the anti-CAMP1 antibody in all *C. acnes* phylotype IA1 strains (*n* = 12) except strain 6919. All these strains contained a CAMP1 protein recognized by TLR-2. CAMP1 was identified with the anti-CAMP1 antibody in six of the nine phylotype IB strains (14230, 12513, 27387, 25236, GUE, PIE), with a variable intensity of binding to TLR-2. The IA2 phylotype strain (CHR) displayed strong CAMP1 expression, with strong recognition by TLR-2. Finally, CAMP1 was detected in the phylotype II strain (27647), but was not recognized by TLR-2. Interestingly, the anti-CAMP1 antibody detected a single protein band at about 27 kDa, or this band together with a second band at about 24 kDa. We investigated the differential expression of CAMP1 in *C. acnes* strains grown on solid media. We showed that the CAMP1 protein was detected as a single band at about 27 kDa when *C. acnes* was grown onto solid media (Figure 1B), whereas the two protein bands were detected after growth in liquid media (Figure 1A). Moreover, we showed that *C. acnes* grown in liquid media displayed a strong induction of CXCL8/IL-8 production (Figure 1D). We, therefore, performed all subsequent experiments with *C. acnes* strains grown in liquid media. Thus, *C. acnes* strains of all phylotypes tested were able to produce CAMP1 protein, and the nature of the CAMP1 protein produced seemed to depend on the growth conditions.

### 2.2. Anti-CAMP1 Antibodies Decrease C. acnes-Induced CXCL8/IL-8 Production In Vitro

We investigated the effect on CXCL8/IL-8 production of the CAMP1 produced by *C. acnes* by performing blocking experiments with an anti-CAMP1 antibody. We used an antibody against TLR-2 as a positive control in the *C. acnes*-induced CXCL8/IL-8 production model in vitro (Figure 2). We showed that this antibody decreased the production of CXCL8/IL-8 in one phylotype IA1 strain and two phylotype IB strains of *C. acnes*, by 87, 81 and 91%, respectively (Figure 2A). In experiments with the anti-CAMP1 antibody and phylotype IB strains producing various amounts of CAMP1, we observed a 48% decrease in CXCL8/IL-8 with the PIE strain, which produces large amounts of CAMP1, but only a 21% decrease with the 14230 strain, which produces small amounts of CAMP1; however, CXCL8/IL-8 levels were higher in the phylotype IA1 strain 6919, which does not produce CAMP1 (Figure 2B). These results suggest that TLR-2 plays a major role in the production of CXCL8/IL-8 by keratinocytes stimulated by *C. acnes* and that the CAMP1 protein may also be involved in this process.

### 2.3. Recombinant CAMP1 Proteins Are Specifically Recognized by TLR-2

We investigated the interaction between CAMP1 and TLR-2, by producing recombinant CAMP1 protein (rCAMP1) from *C. acnes* strain group A expressing non-mutated CAMP1 (nmrCAMP1) and from *C. acnes* strain group F expressing mutated CAMP1 (mrCAMP1) [24] (Appendix A). Recombinant CAMP1 proteins were separated by electrophoresis and subjected to far-Western blotting for TLR-2, TLR-1, TLR-4 and TLR-6, as described in the Materials and Methods. We found that mrCAMP1 and nmrCAMP1 (47 kDa) were recognized by TLR-2 (Figure 3A) but not by TLR-1, TLR-4 and TLR-6. Interestingly, differences in TLR-2 binding intensity were observed, with nmrCAMP1 recognized less strongly than mrCAMP1. TLR-2 is also bound to a protein at about 60 kDa and also observed in the control experiment with the secondary antibody alone. Another protein, at 30 kDa, was also recognized strongly by TLR-2 and less strongly by TLR-1. We investigated the nature of this protein, by excising the bands from the gel after electrophoresis and subjecting them to proteolysis and mass spectrometry (MS)/MS analysis. A similar analysis was performed for the rCAMP1 band at 47 kDa (Figure 3B). We confirmed the sequences of mrCAMP1 and nmrCAMP1, with 53% and 57% coverage, respectively. The 30 kDa protein appears to be a fragment of CAMP1 and was sequenced with 33% coverage. This 30 kDa CAMP1-related protein was recognized by TLR-2 and by antibodies against the poly-His tag. For confirmation of binding results, we then immobilized mrCAMP1 and nmrCAMP1 on plates, which we then probed with various concentrations of TLRs. We observed strong, dose-dependent and saturable binding to TLR-2 for both nmrCAMP1 and mrCAMP1. The binding intensity was lower for nmrCAMP1, for which an attenuation of binding tended to occur at high TLR-2 concentrations, although this difference was not statistically significant. In parallel, weak binding activity was detected with TLR-4 and TLR-1, and no binding activity was detected with TLR-6 (Figure 3C). These results indicate that the interaction between CAMP1 and TLR-2 is specific.

### 2.4. Identification of TLR-2 Binding Motifs in CAMP1 and In Silico 3D-Structure Modeling

We investigated the CAMP1 peptide sequences involved in binding to TLR-2 by synthesizing a total of 116, 15-amino acid (aa) peptides overlapping by 11 aa and with a gap of 4 aa (Figure 4, Appendix A). These peptides were immobilized on a glass slide (spots A1 to E20) and their binding to TLR-2 was then assessed. Control experiments were performed in parallel with the secondary antibody against TLR-2 alone, to eliminate non-specific recognition. Several peptide spots (A20, A21, B18, B19, B20, C5, C6, C7, C8, C9, C17, D8, D9, D11, D12, D13, D14, D19, D20, D22, D23, D24, E3, E4, E5, E6, E17, E18, E19) were recognized by both the TLR-2 and the control antibody, indicating a non-specific interaction (Figure 4A,B); however, seven peptides appeared to be recognized specifically by TLR-2 (A14, A15, B1, B2, B3, C1, C3) (Table 1). We investigated the role of these seven peptides in binding to TLR-2, by synthesizing them separately and assessing their ability to inhibit CAMP1–TLR-2 binding in a quantitative assay (Figure 4D). We found that peptides A14 and B2 decreased CAMP1-TLR-2 binding activity in a dose-dependent manner, whereas peptides A15, B1 and C3 decreased this binding activity only at higher concentrations. Finally, neither B3 nor C1 decreased CAMP1-TLR-2 binding activity. These results suggest that the CAMP1 peptide sequences involved in TLR-2 binding correspond to the A14, A15, B1, B2 and C3 peptides.

We performed a three-dimensional (3D) modeling analysis to localize these peptide sequences within the CAMP1 protein. A comparison of *C. acnes* nm- and mCAMP1 sequences with the CAMP sequences from *S. agalactiae*, *M. curtisii* and *S. pyogenes* revealed sequence identities of 28%, 30% and 28%, and similarities of 46%, 45% and 47%, respectively (Figure 5A). Based on this identity analysis, we performed an in silico 3D-structure analysis of the *C. acnes* CAMP1 by the homology modeling method, as previously described [25]. We identified two domains: the N-terminal domain (NTD), composed of five helices, and the C-terminal domain (CTD), composed of three helices. These two domains are linked by a small linker region. No major structural differences were found between m- and nmCAMP1 (Figure 5B). Interestingly, the CAMP1 peptides involved in TLR-2 binding were mostly localized to the NTD domain on one side of the 3dimensional (3D)-CAMP1 structure (Figure 5C). Moreover, a peptide surface representation based on the area of water molecules in contact with the protein at all possible positions showed that all the peptides were located close together and formed a pocket-like structure in which interactions with TLR-2 could occur (Figure 5D). Thus, our results suggest that the CAMP1 peptides involved in TLR-2 binding are all located on one side of the CAMP1 structure, forming what appears to be a binding pocket.

### 2.5. CAMP1-Derived Peptides Decrease C. acnes-Induced CXCL8/IL-8 Production In Vitro

CXCL8/IL-8 production involves the TLR-2 signaling pathway [26]. We, therefore, investigated the effects of the identified peptides on CXCL8/IL-8 production in keratinocytes. HaCaT keratinocytes were first treated with various concentrations (3.9 to 250 μM) of peptides (A14, A15, B1, B2, B3, C1, C3) and were then stimulated with *C. acnes*. A14, B1 and B2 significantly decreased CXCL8/IL-8, in a dose-dependent manner, with an IC_50_ of about 30–60 μM, whereas A15, B3, C1 and C3 did not (Figure 6A). We confirmed the effect of the A14, B1 and B2 peptides, at a concentration of 62.5 μM, by demonstrating a downregulation of CXCL8/IL-8 mRNA and protein levels (Figure 6B,C). Peptides A14 and B1 decreased cell viability by 30% at the highest concentrations (Figure 6D). Peptides A15, B2, B3 and C1 displayed no significant toxicological activity, and C3 was toxic only at the highest concentration. The A14, B1 and B2 peptides are located on the same side in the 3D CAMP1 structure (Figure 5C). 

We evaluated the effects of rCAMP1 on CXCL8/IL-8 production. HaCaT keratinocytes cells were first incubated with various concentrations of nmrCAMP1. This protein stimulated the production of CXCL8/IL-8 by HaCaT keratinocytes in a dose-dependent and saturable manner to levels similar to those recorded for peptidoglycan (PGN), which was used as a positive control (Figure 6E). A comparison of nm- and m rCAMP1 showed that both induced CXCL8/IL-8 production, but that mrCAMP1 had a stronger effect on both mRNA and protein levels (Figure 6F,G). Thus, both rCAMP1 proteins can be considered potent inducers of CXCL8/IL-8 in vitro. The prior treatment of primary keratinocytes with A14, B1 or B2 decreased CXCL8/IL-8 production by 62, 74 and 54%, respectively (Figure 6H). Thus, specific CAMP1-related peptides can decrease CXCL8/IL-8 production in vitro.

### 2.6. CAMP1-Derived Peptides Decrease Proinflammatory Molecule Production Ex Vivo

Human skin explants were first incubated with various amounts of *C. acnes* suspension (O.D. at 620 nm of 0.3 to 2). TNF-α production, used as first read out during the experiment, was found to be dose-dependent, reaching up to 64 pg/mL (*p* < 0.0001) but decreasing at high bacterial loads. For subsequent experiments, we, therefore, used bacteria at an OD_620 nm_ of 1.0, corresponding to about 7 x 10^7^ bacteria/mL (Appendix A). For validation of this ex vivo model of *C. acnes*-induced inflammation, skin explants were incubated with lipopolysaccharide (LPS) or peptidoglycan (PGN) as positive controls, and with *C. acnes* strains of phylotypes IA1, IA2 and II. This stimulation resulted in significantly higher levels of TNF-α production (22 ± 1.7, 126 ± 9.3, 46 ± 3.9, 44 ± 2.2 and 42 ± 1.9 pg/mL, for LPS, PGN, and phylotypes IA1, IA2 and II, respectively) than for unstimulated explants. GM-CSF levels were 765 ± 30, 4733 ± 218, 1373 ± 118, 1410 ± 55 and 2731 ± 102 pg/mL, respectively. IL-1β levels were 392 ± 5.6, 23 ± 0.8, 79 ± 3.5, 49 ± 1.0 and 60 ± 1.0 pg/mL, respectively. A smaller induction of IL-10 production, to levels of 10 ± 0.3, 35 ± 1.4, 15 ± 0.5, 14 ± 0.5 and 21 ± 0.8 pg/mL, respectively, was also observed. IL-1α and IL-17α levels were very low. We detected no matrix metlloproteinase (MMP)-1 or MMP-3 after stimulation with the *C. acnes* strains (Appendix A). The levels of CXCL8/IL-8 were very high due to the presence of SVF in the culture medium, and were not considered informative (Appendix A). As stimulation with *C. acnes* strains resulted in the production of proinflammatory molecules, we then tested the effect of the CAMP1-related peptides A14, B1 and B2 in this model, following stimulation with both nmrCAMP1 (Figure 7) and *C. acnes* strains (Appendix A). Human skin explants were first treated with the A14, B1 and B2 peptides (62.5 μM) and were then stimulated by incubation with the nmrCAMP1 protein at a concentration of 10 μg/mL. As previously observed with *C. acnes* strains, the rCAMP1 protein induced the production of GM-CSF (2582 pg/mL, *p* < 0.0001), IL-1β (14.9 pg/mL, *p* < 0.0001), TNF-α (62.7 pg/mL, *p* < 0.0001) and IL-10 (21.3 pg/mL, *p* < 0.0001); however, much larger amounts of MMP-1 (5965 pg/mL, *p* < 0.0001) and MMP-3 (32431 pg/mL, *p* < 0.0001) were produced than with unstimulated skin explants (Figure 7E,F). In the presence of the A14, B1 and B2 peptides, GM-CSF levels decreased by 97% (*p* < 0.0001), 92% (*p* < 0.0001) and 93% (*p* < 0.0001), respectively. IL-1β levels decreased by 77% (*p* < 0.0001), 77% (*p* < 0.0001) and 63% (*p* < 0.00001), respectively. TNF-α decreased by 88% (*p* < 0.00001), 88% (*p* < 0.00001) and 83% (*p* < 0.00001), respectively. IL-10 levels decreased by 85% (*p* < 0.0001), 86% (*p* < 0.0001) and 77% (*p* < 0.0001), respectively. MMP-1 levels decreased by 68% (*p* = 0.0332), 51% (*p* = 0.0885) and 49% (*p* = 0.0289), respectively. MMP-3 levels decreased by 44% (*p* = 0.0243), 42% (*p* = 0.0361) and 54% (*p* = 0.0006), respectively.

Similar results were obtained for skin explants treated with the A14, B1 and B2 peptides (62.5 μM) and then stimulated with *C. acnes* (Appendix A). Peptide pre-treatment decreased GM-CSF mRNA levels by 99.9% (*p* = 0.0027), 99.8% (*p* = 0.003) and 99.8% (*p* = 0.0027), respectively. For IL-1β mRNA, the decrease was 97% (*p* = 0.0004), 97% (*p* = 0.0005) and 97% (*p* = 0.0005), respectively. For TNF-α mRNA, the decrease was 70% (*p* = 0.0009), 50% (*p* = 0.0445) and 45% (*p* = 0.170, non-significant), respectively. For IL-10 mRNA, the decrease was 99% (*p* = 0.0027), 99% (*p* = 0.0027) and 99% (*p* = 0.0027), respectively (Appendix A). Similar effects were observed for protein levels for GM-CSF (3751 pg/mL, *p* < 0.0001), IL-1β (19.4 pg/mL, *p* < 0.0001), TNF-α (3.8 pg/mL, *p* < 0.0001) and IL-10 (7.6 pg/mL, *p* < 0.0001). Relative to skin explant stimulation with *C. acnes* alone, pre-treatment with peptides A14, B1 and B2, decreased the level of GM-CSF by 91% (*p* < 0.0001), 90% (*p* < 0.00001) and 89.9% (*p* < 0.0001), respectively. IL-1β levels decreased by 79.5% (*p* < 0.0001), 57.2% (*p* < 0.0001) and 68% (*p* < 0.0001), respectively. TNF-α levels decreased by 70% (*p* = 0.0049), 67.5% (*p* = 0.0092) and 74.9% (*p* = 0.0024), respectively. IL-10 levels decreased by 69.5% (*p* < 0.0001), 74% (*p* < 0.0001) and 68% (*p* < 0.0001), respectively (Appendix A). Western-blot analysis showed that both *C. acnes* and rCAMP1 proteins activated the phosphorylation of extracellular signal-related kinase (ERK) and p38 mitogen-activated kinase (p38) in skin explants. Pretreatment with the A14, B1 and B2 peptides followed by nmrCAMP1 stimulation inhibited p38 phosphorylation, and, to a lesser extent, that of ERK. The blot was stripped and re-probed with antibodies against total ERK and p38, demonstrating an absence of change in total protein levels following *C. acnes* stimulation (Figure 8A,B). Thus, the CAMP1-related peptides A14, B1 and B2 significantly decreased the production of proinflammatory molecules via the MAPK pathway.

### 2.7. CAMP1-Related Peptide Pretreatment Downregulates Inflammation-Related Genes

We assessed the effect of the B2 peptide on genome-wide gene expression by next-generation sequencing (NGS) of mRNA at a depth of approximately 30 million mapped reads per skin explant treated with B2 and stimulated with nmrCAMP1. STAR v2.7.6a was used to align all the reads obtained with the human transcriptome for the GRCh38.101 human reference genome. Crude counts were normalized and a score for aligned reads per transcript per million reads was calculated (RSEM V1.3.1). These scores were used to differences in expression, which were considered significant for a ± 1.5-fold change. A cluster pattern analysis was performed on mRNAs differentially expressed between the unstimulated and untreated explant group (negative control group, *n* = 3), the nmrCAMP1-stimulated explant group (CAMP1-stimulated group, *n* = 3) and the group of skin explants treated with B2 and then stimulated with nmrCAMP1 (B2-treated group, *n* = 2). In total, 19,470 differentially expressed genes were subjected to filtering with thresholds of *p* < 0.05 and log_2_(fold-change) of at least 1.5 in either direction (Figure 9A). A comparison between the positive control group and the treated group showed that 275 genes were upregulated and 963 were downregulated by the treatment (Figure 9B). Kyoto encyclopedia of genes and genomes (KEGG) pathway enrichment analysis with a cutoff of *p* < 0.05, log2(fold-change) of at least 1.5 in the negative direction and basal level mean >100 for the B2-treated group identified 443 genes as downregulated. These genes were distributed between 56 pathways, 41 of which were statistically significant (*p* < 0.05). The 20 most remarkable pathways are presented in Figure 9D. A comparison between the negative control group and the CAMP1-stimulated group based on the same cutoff criteria showed that 661 genes were upregulated and 222 genes were downregulated (Figure 9C). KEGG pathway enrichment analysis identified 267 genes as upregulated. These genes were distributed between 58 pathways, 53 of which were statistically significant (*p* < 0.05), and the 20 most remarkable pathways are presented in Figure 9E (Appendix A). Most of the pathways identified in this analysis were related to the inflammatory response. We identified 12 pathways common to both sets of conditions (cytokine–cytokine receptor interaction, IL-17 signaling pathway, TNF signaling pathway, MAPK signaling pathway, NF-kappa B signaling pathway, hematopoietic cell lineage, viral protein interaction with cytokine and cytokine receptor, nucleotide-binding oligomerization domain (NOD)-like receptor signaling pathway, C-type lectin receptor signaling pathway, FoxO signaling pathway, cellular senescence and chemokine signaling pathway). The 10 most significant differentially expressed genes (DEGs) from these pathways were *IL6*, *TNF*, *CXCL1*, *CXCL2*, *CXCL3*, *CXCL8*, *IL-1β*, chemokine ligand (*CCL*)*2*, *CCL4* and colony stimulating factor (*CSF*)*2* (Table 2). The results obtained for the *TNF*, *CXCL8*, *IL1B* and *CSF2* genes were consistent with those described above. Moreover, a separate experiment confirmed the pattern of transcriptional regulation for *IL6*, *CXCL1*, *CXCL2*, *CXCL3*, *CCL3* and *CCL4* (Appendix A).

### 2.8. Bacterial Growth Is Not Altered by a CAMP1-Related Peptide 

We investigated the effect of the B2 peptide on *C. acnes*, *S. aureus* and *S. epidermidis*, by adding the peptide to the medium at concentrations of 1 to 128 μM and monitoring bacterial growth (Figure 10). As a positive control for growth inhibition, we cultured the bacteria in the presence of the antibiotic ampicillin, doxycycline and clindamycin. *C. acnes* and *S. aureus* were susceptible to all three antibiotics, whereas *S. epidermidis* was susceptible to doxycycline and clindamycin but resistant to ampicillin (Figure 10A–C). The B2 peptide had no impact on the growth of *C. acnes*, *S. aureus* and *S. epidermidis* (Figure 10D).

## 3. Discussion

Christie–Atkins–Munch–Petersen (CAMP) factors are proteins that can form pores in host membranes, contributing to tissue damage. They were discovered through a synergic interaction between a CAMP factor from *Streptococcus agalactiae* and the β-toxin (sphingomyelinase or SMase) of *Staphylococcus aureus*, inducing the formation of pores in sheep erythrocyte membranes, leading to the lysis of these cells [27]. SMases can hydrolyze sphingomyelin, a component of the erythrocyte membrane. The membrane is vulnerable if it has a sphingomyelin content of at least 45%; rabbit, mouse and human erythrocytes are therefore not susceptible to CAMP factors, whereas goat, sheep and cow erythrocytes are [28].

An analysis of the *C. acnes* genome has revealed the presence of five CAMP factor genes (*camp1*, *camp2*, *camp3*, *camp4* and *camp5*), all considered to be putative virulence factors [29]. The hemolytic properties of *C. acnes* in situ in cases of acne remain unclear, but it has been shown that *C. acnes* can use the host SMase to enhance its virulence [30], and that *C. acnes* strains displaying hemolytic activity have been isolated from the intervertebral discs of patients with chronic lower back pain [31]. *C. acnes* CAMP factor 2 is considered to be a virulence factor because it can induce inflammation in vivo and has been used as an antigen to elicit monoclonal antibodies, decreasing the production of CXCL8/IL-8 and IL-1β in ex vivo skin explants from patients with acne [32,33]. By contrast, the role of CAMP1 remains unknown. However, CAMP1 is mostly expressed by *C. acnes* in the PSU [34], suggesting a role in triggering inflammation. 

We previously showed that *C. acnes* phylotype IA1 strains contain a CAMP1 protein that is only poorly recognized by TLR2, if at all [24]. In this study, using polyclonal antibodies against CAMP1, we showed that CAMP1 was present in *C. acnes* strains of phylotype IA1. We also detected CAMP1 in *C. acnes* phylotypes IA2 and IB, consistent with the findings of a previous study [35]. Our results conflict with those of Lheure et al. [24] for phylotype II, in which we were unable to detect CAMP1 binding to TLR2, despite performing several experiments; however, a larger number of *C. acnes* strains of phylotypes IA2, II and III will need to be analyzed, to evaluate the expression of CAMP1 in this species as a whole. CAMP1-TLR-2 binding activity assays detected CAMP1 in phylotypes IA2 and IB, as in our previous study [24]. Anti-CAMP1 antibodies detected the presence of a single protein band at about 27 kDa, or two protein bands at about 24 and 27 kDa, as previously described in a study using TLR-2 as the probe [24]. We investigated whether growth conditions affected CAMP1 production, by growing *C. acnes* strains on a solid medium. Surprisingly, CAMP1 was detected as a single 27 kDa band in these conditions, regardless of phylotype. Moreover, *C. acnes* seemed to induce higher levels of CXCL8/IL-8 production when grown in a liquid medium, suggesting that growth conditions might influence CAMP1 expression and the ability of strains to induce chemokine production. This finding is consistent with a previous study reporting the differential expression of surface proteins according to the culture medium [36]. With the anti-CAMP1 antibody, we showed that keratinocytes stimulated with *C. acnes* phylotype IB, which expresses a CAMP1 recognized by TLR-2, produced less CXCL8/IL-8 production than those stimulated with other phylotypes. We found that higher levels of CAMP1 in a *C. acnes* strain were associated with a stronger effect of the anti-CAMP1 antibody. Conversely, stimulation with *C. acnes* phylotype IA1, which produces a CAMP1 that is poorly recognized by TLR-2, if at all, resulted in no decrease in CXCL8/IL-8 production. Control experiments with antibodies against TLR-2 showed that CXCL8/IL-8 production decreased with all phylotypes tested, as in previous studies [14,24]. Overall, these results indicate that CAMP1 is expressed in phylotypes IA1, IA2 and IB, and that the CAMP1 protein recognized by TLR-2 is involved in CXCL8/IL-8 expression in keratinocytes; however, more investigations and the generation of Δ*camp1*
*C. acnes* strains will be required to determine the impact of CAMP1 on chemokine production.

We evaluated the binding of CAMP1 to TLR-2 by producing recombinant proteins from *C. acnes* strains expressing non-mutated and mutated CAMP1. Far-Western blotting experiments showed that rCAMP1 was recognized by TLR-2, but not by TLR-1, TLR-4 and TLR-6. This result is consistent with a previous study in which a *C. acnes* surface protein was identified as CAMP1 [24]. The quantitative difference in TLR-2 binding to nmrCAMP1 and mrCAMP1 was not significant, but the intensity of the signal was lower for nmrCAMP1. This result is consistent with a previous study in which *C. acnes* strains expressing an unmutated CAMP1 protein displayed weaker TLR-2 binding [24]. We detected a band at about 60 kDa that was recognized by TLR-2. This protein was also recognized by the anti-TLR2 secondary antibody, so the reaction was considered non-specific. We also detected a protein at about 30 kDa that was recognized strongly by TLR-2, more weakly by TLR-1 and not all by TLR-4 and TLR-6. This 30 kDa protein was recognized by the His-Tag antibody and proteomic analysis showed it to be a component of the CAMP1 protein. This result suggests possible differential splicing of rCAMP1 during its synthesis in *E. coli*. The quantification of rCAMP1-TLR binding activities showed that recognition by TLR-2 was dose-dependent and saturable, consistent with a specific binding process. TLR-1 and TLR-4 displayed very moderate binding activities, whereas no binding was detected with TLR-6. This result is consistent with the far-Western blotting experiment and indicates that CAMP1-TLR-2 recognition is specific.

We investigated the CAMP1-TLR-2 interaction further by using a peptide spot array library of the CAMP1 protein sequence to identify the parts of the molecule involved in binding to TLR-2 binding. This method made it possible to screen a large number of peptides, as previously described [37]. Seven of the 15-mer peptides were specifically recognized by TLR-2. We eliminated the C17 peptide, because it also gave a high level of binding with the control, indicating that the interaction was non-specific. Moreover, the C17 peptide did not belong to the same group as the peptides surrounding its amino acid sequence. Indeed, the selected peptides belonged to three groups (A14-A15, B1-B2-B3, C1-C3) corresponding to overlapping sequences, consistent with a role for a consensus amino acid sequence in binding; however, a competitive binding inhibition experiment showed that A14 and B2 partially inhibited CAMP1-TLR-2 binding in a dose-dependent manner, whereas the other peptides either failed to inhibit this binding or did so only at the highest concentration. The discrepancy between the peptide array and plate inhibition assay results may reflect differences in peptide spatial presentation, as the peptide is immobilized on cellulose in the array and in solution during the inhibition assay on plates. These results suggest that A14 and B2 have the best binding inhibition capacities of the CAMP1 peptides recognized by TLR-2.

We were unable to crystallize rCAMP1 in this study because its purity was not sufficiently high; however, 3D-crystal structure analyses have already been performed on CAMP factors from *Mobiluncus curtisii* and *Streptococcus agalactiae* [38,39], and a comparison between the sequences of the *S. agalactiae* CAMP protein and the *C. acnes* CAMP1 protein revealed a sufficient level of sequence identity for in silico 3D structure analysis [25]. The 3D structure of *C. acnes* CAMP1 appears to be similar to that of the *S. agalactiae* CAMP protein, with N-terminal and C-terminal domains separated by a linker region. The peptides recognizing TLR-2 were found to be located on the same side of the structure, with the A14 and B2 close to each other spatially. TLR-2 belongs to the Toll-IL-1 receptor superfamily. The receptors of this family have an intracellular Toll interleukin-1 receptor domain (TIR) responsible for signaling pathway activation and an extracellular leucine-rich repeat domain (LRR) responsible for the interaction with lipopeptides. TLR-2 forms heterodimers with TLR-1 and TLR-6, which in turn, recognize di- and tri-acylated lipoproteins via the hydrophobic pocket [40]; however, several studies have shown that proteins produced by *Neisseria meningitidis* or *Vibrio cholerae* can also activate TLR-2 to induce proinflammatory cytokine responses in a TLR2-dependent manner [41,42]. Polysaccharides also seem to be able to interact with TLR-2 [43]. Moreover, a small molecule, diprovocim, with a structure very different from the canonical lipostructures recognized by TLR-2, was able to bind to the ligand-binding pocket of the heterodimerized LRR of TLR-2 [44]. No crystal structure showing the mechanism of TLR-2 binding to proteinaceous ligands has yet been obtained. Further studies will be required to crystallize the *C. acnes* CAMP1 for the determination of its 3D structure and investigation of the CAMP1-TLR-2 interaction.

*C. acnes* can induce high levels of chemokine and cytokine production by the various target cells it encounters in the course of inflammatory acne. We showed that both the nmrCAMP1 and mrCAMP1 proteins induce a strong proinflammatory molecule response in HaCaT and primary keratinocytes. This result confirms previous findings that CAMP1 proteins eluted from electrophoretic gels can induce CXCL8/IL-8 production by activating the TLR-2 signaling pathway [24]. The A14, B1 and B2 peptides yielded a dose-dependent decrease in CXCL8/IL-8 production, whereas the other peptides recognized by TLR-2 (A15, B3, C1, C3) did not, suggesting that a specific amino acid sequence and/or conformation is involved in this inhibition. No major impact on cell viability was observed, indicating that the decrease in CXCL8/IL-8 production did not result from cell death. Subsequent investigations focused on peptides A14, B1 and B2, which were the only ones with a significant effect on CXCL8/IL-8 production. We used an ex vivo model of *C. acnes*-induced inflammation based on human skin explants from healthy individuals. The stimulation of these explants with *C. acnes* strains led to the massive production of TNF-α, GM-CSF and IL-1β. This result is consistent with previous reports of the expression of these cytokines in lesional skin from acne patients [33,45]; however, metalloproteinase expression was weak following stimulation with *C. acnes*. This discrepancy may be due to the use of normal healthy skin explants in our study and lesional skin explants from acne patients in previous reports. We detected the production of the anti-inflammatory IL-10 cytokine, but at levels far lower than for TNF-α, GM-CSF and IL-1β. This result is consistent with previous findings showing IL-10 production in skin explants from acne lesions and from healthy individuals [46]. IL-17 expression was weak in this model. This result is consistent with previous reports that *C. acnes* can induce the production of IL-17 by PBMCs and in acne lesions, indicating the presence of a CD4^+^ T-cell infiltrate. The low level of IL-17 in our model may be due to the low counts of CD4^+^ cells and CD83^+^ mature dendritic cells in non-lesional skin, as previously reported [45,47]. Similarly, only very small amounts of IL-1α were produced, whereas this cytokine has been reported to be produced in large amounts in keratinocytes in vitro and in a mouse model in vivo involving a sebum-like application of acne-related *C. acnes* strains [48,49]. CXCL8/IL-8 levels were very high in the skin explant culture medium due to the presence of SVF in the medium. We, therefore, considered the values obtained for CXCL8/IL-8 to be uninformative in this model. We found no major difference between the *C. acnes* phylotypes in terms of the levels of proinflammatory molecules produced, but only one strain per phylotype was tested. The investigation of larger numbers of *C. acnes* strains is therefore required for any firm conclusions to be drawn. Moreover, comparisons between healthy- and acne-related *C. acnes* strains are likely to be of interest. This model is not directly comparable to acne lesions, but it could, nevertheless, be useful for investigations of the *C. acnes*-induced response, as the skin explant may contain some of the cells involved in the immune response.

We then assessed the impact of peptides A14, B1 and B2 on skin explants stimulated with both *C. acnes* and nmrCAMP1. These explants displayed significantly lower levels of TNF-α, GM-CSF and IL-1β production and also had significantly lower levels of the metalloproteinases MMP1 and MMP3. Metalloproteinases are enzymes with proteolytic activity that are involved in the control of extracellular matrix (ECM) remodeling [50]. They are induced by reactive oxygen species (ROS) produced in response to several stresses, including UV exposure, TNF-α [51,52] and *C. acnes* infection [53]. The ROS activate the MAPK pathway, thereby activating the transcription factor activator protein-1 (AP-1), leading to the MMP expression [54]. MMPs play a key role in acne scarring and *C. acnes* upregulates several of these enzymes [55]. CAMP1-derived peptides, therefore, seem to be able to downregulate all the major inflammatory components induced by *C. acnes*. We further investigated the impact of the B2 peptide on the inflammatory response induced by *C. acnes*, by performing an RNA sequencing analysis on skin explants treated with the B2 peptide and stimulated with nmrCAMP1. Most of the inflammatory markers strongly upregulated in the presence of nmrCAMP1 were downregulated in the presence of B2. We identified 12 pathways significantly affected in both untreated and treated conditions: the cytokine–cytokine receptor interaction, the IL-17 signaling pathway, the TNF signaling pathway, the MAPK signaling pathway, the NF-κB signaling pathway, the hematopoietic cell lineage, the viral protein interaction with cytokine and cytokine receptor, the NOD-like receptor signaling pathway, the C-type lectin receptor signaling pathway, the forkhead (Fox)O signaling pathway, cellular senescence and the chemokine signaling pathway. This finding is consistent with previous reports identifying the cytokine–cytokine receptor interaction pathway as active in acne lesions, supporting the notion that this model is a potentially interesting tool for studies of the innate immune molecular pathways induced by *C. acnes* [56,57]. We identified 10 DEGs from these pathways: *IL6*, *TNF*, *CXCL1*, *CXCL2*, CXCL3, CXCL8, IL-1β, CCL2, CCL4 and CSF2. All these genes encode proinflammatory cytokines and chemokines previously detected in acne lesions [58] and are linked to activation of the NF-κB pathway, inducing the upregulation of TNF-α, CXCL8/IL-8, IL-1β, CXCL1 and CXCL2 [16,59,60]. The B2 peptide decreases the inflammatory response induced by *C. acnes* and can therefore be considered a novel modulator of the TLR-2 signaling pathway. The use of an ex vivo human skin explant model also appears to be a useful tool for studies of the impact of candidate anti-inflammatory molecules.

The cutaneous microbiota has been shown to play an essential role in maintaining skin homeostasis and health, with imbalances leading to several diseases. *C. acnes* is by far the most abundant of the cutaneous bacteria and the selection of a specific acne-associated lineage may be linked to skin dysbiosis [61,62]. The increase in antibiotic resistance in *C. acnes* and other skin-related bacteria is making it necessary to develop and test new therapeutic approaches. Peptide-based approaches, in particular, are being considered; however, this technology is based essentially on disruption of the membrane to decrease bacterial viability and the associated inflammatory reaction [63,64,65]. We show here that the CAMP1-related B2 peptide did not alter the growth of *C. acnes*, *S. aureus* and *S. epidermidis* strains at high concentrations. It has been shown that TLR-2 modulators can be used to treat acne without adverse events [66]. The B2 peptide derived from the *C. acnes* CAMP1 protein, which has strong anti-inflammatory properties, is, therefore, a good candidate for clinical testing.

## 4. Materials and Methods

### 4.1. Bacterial Strain and Conditions of Growth

We used 24 strains of *C. acnes* in the study: strain 6919 (Type IA_1_) from the American Type Culture Collection (Manassas, VA) and strains isolated from various infections (13 bone and joint infections, 1 case of pleuritis, 1 case of appendicitis, 1 case of pericarditis, 1 case of pneumonia, 3 cutaneous infections, 1 case of bacteremia and 2 ascitic fluid infections). *C. acnes* strains RON (IA_1_), TRI (IA_2_), LIE (IA_2_), CHR (IA_2_), GUE (IB) and PIE (IB) were isolated from patients with joint infections. All these strains were kindly provided by Dr. Philippe Morand from the Bacteriology Department at Cochin Hospital, Paris, France [24]. *C. acnes* strains were grown under anaerobic conditions in liquid or solidified reinforced clostridial medium (RCM; Difco Laboratories, Detroit, MI, USA), in a GasPak™ EZ Anaerobic Container System (Becton Dickinson & Co, Franklin Lakes, NJ, USA) at 37 °C. For routine culture, 100 mL of RCM was used and the bacteria were harvested after 5 days at 37 °C, by centrifugation at 7,000 x *g* for 10 min and 4 °C. The pellets were pooled and washed in about 30 mL cold sterile phosphate-buffered saline (PBS) (1.5 mM KH_2_PO_4_, 2.7 mM Na_2_HPO_4_.7H_2_O, 0.15 M NaCl (pH 7.4)), and were then centrifuged again, as described above. The final bacterial pellet was suspended in sterile PBS (1:10 based on the volume of the culture). *Staphylococcus aureus* (ATCC #25923) and *Staphylococcus epidermidis* (ATCC #12228) were grown in tryptic soy broth (TSB) and brain heart infusion (BHI) (Sigma-Aldrich, Saint-Louis, MO, USA) under aerobic conditions at 37 °C.

Two chemically competent *Escherichia coli* strains (TOP10 and BL21 Star^TM^ DE3; Thermo Fisher Scientific, Life Technologies SAS, Courtaboeuf, France) were used. The strains were grown under aerobic conditions at 37 °C in liquid or solidified Luria-Bertani (LB) (Sigma-Aldrich, Saint-Louis, MO, USA) medium with or without ampicillin (50 μg/mL) to screen for transformants. Super optimal broth with catabolite repressor (SOC medium) was used for cell recovery after heat shock during the transformation experiments.

### 4.2. Total Bacterial Surface Protein Extraction

The *C. acnes* total protein extract was obtained by incubating the bacterial pellet with chicken egg lysozyme (2 mg/mL) for 1 h at 37 °C, with gentle mixing. The suspension was centrifuged at 10,000× *g* for 15 min at 4 °C and the supernatant containing the proteins was retained. Protein concentration was determined by the Lowry method, with BSA as the standard [67].

### 4.3. Production of Polyclonal anti-CAMP1 Antibodies

Antibodies directed against CAMP1 (accession number AAS92206.1) were obtained by conjugating two peptides with keyhole limpet hemocyanin (carrier protein used as a vaccine adjuvant): KMPDLKPNDVAT and VLRQIRFDRNTC. The resulting conjugates were injected into female New Zealand White rabbits (Proteogenix, Schiltigheim, France). Three injections were performed over a period of 70 days. Blood was obtained from the rabbits at the end of the immunization protocol. Antisera were first depleted against immobilized non-phosphorylated peptide, and specific antibodies against CAMP1 were then purified with the phosphorylated KMPDLKPNDVAT and VLRQIRFDRNTC peptides immobilized on a column. The titer of purified antibodies was evaluated by enzyme-linked immunosorbent assay (ELISA) with the phosphorylated and non-phosphorylated peptides and was found to be 1:64,000 (Appendix A).

### 4.4. Far-Western Blotting

*C. acnes* surface proteins (50 μg) were separated by electrophoresis under denaturing conditions in a NuPAGE Novex 4–12% Bis-Tris gel (Thermo Fisher Scientific, Life Technologies SAS, Courtaboeuf, France). The separated proteins were transferred onto a nitrocellulose membrane, which was then incubated in saturation buffer containing 1 X tris-buffered saline (TBS), 1.4 M NaCl, 5% non-fat milk, 0.1% Tween 20, 200 mM Tris (pH 7.6) for 1 h at room temperature. The membrane was washed three times, for 15 min each, with TBS/T buffer (1 X TBS, 0.1% Tween-20) and were then incubated overnight with human recombinant TLR-1, TLR-2, TLR-4 or TLR-6 (R&D Systems, Abingdon, UK) diluted to 0.1 μg/mL in TBS/T, at 4 °C, with gentle shaking. The unbound protein probe was removed by washing as described above, and the membranes were incubated with biotinylated human antibodies against TLR-1, TLR-2, TLR-4 or TLR-6 (R&D System, Abingdon, UK) diluted to 0.05 μg/mL in TBS/T supplemented with 5% BSA, for 20 h at 4 °C, with gentle shaking. The membranes were washed to remove unbound antibodies, and the surface protein-TLR1,2,4 or 6-biotinylated antibody complexes were detected by incubation for 1 h at room temperature with HRP (horseradish peroxidase) diluted to 0.5 μg/mL in saturation buffer. Unbound material was removed by washing and peroxidase activity was detected in a chemiluminescence assay (WesternBright ECL, Advansta, Menlo Park, CA, USA).

### 4.5. CAMP1 Binding Activity

For quantitative analysis, nmrCAMP1 and mrCAMP1 proteins were diluted to 20 μg/mL in 50 mM carbonate–bicarbonate buffer (pH 9.6) and immobilized on 96-well polystyrene plates by overnight incubation at 4 °C. The wells were washed three times with 0.05% Tween-20 in PBS (PBS-Tween). Human recombinant TLR-1, TLR-2, TLR-4 or TLR-6 (R&D System, Abingdon, UK) (0.01 to 16 μg/mL in PBS-Tween) was added and the plates were incubated overnight at 4 °C. The plates were washed three times with PBS-Tween, and biotinylated human antibodies against TLR-1, TLR-2, TLR-4 or TLR-6 (R&D System, Abingdon, UK) diluted to 0.05 μg/mL in PBS-Tween were added and the plates were incubated at room temperature for 1 h. The plates were washed, peroxidase-conjugated streptavidin (0.5 μg/mL in PBS-Tween) was added and the plates were incubated for 30 min at room temperature. After washing, bound peroxidase activity was detected by incubation with the chromogenic peroxidase substrate ABTS. Absorbance was measured at 405 nm. For the binding inhibition experiments, nmrCAMP1 was immobilized on a plate at 20 μg/mL as described above. In parallel, TLR-2 (0.1 μg/mL) was incubated overnight at 4 °C with various concentrations of the A14, A15, B1, B2, B3, C1 or C3 (1.25, 12.5, 125, 1250 and 12500 nM) peptides, which were added to the plate after it had been washed with PBS-Tween to remove excess nmrCAMP1. Binding was then detected as described above. The peptides (chemically synthesized by Proteomic Solutions, Saint Marcel, France) were dissolved in 30% DMSO for A14 and A15 and in water for B1, B2, B3, C1 and C3.

### 4.6. Cell Culture, Pretreatment and Stimulation

Immortalized human keratinocytes HaCaT (Catalog #300493; GmbH, CLS Cell Lines Services GmbH, Eppelheim, Germany) were grown in Dulbecco’s modified Eagle’s medium-Glutamax-I (DMEM) supplemented with 1 mM sodium pyruvate. The immortalized human monocytic cell line ThP1 (ATCC, Manassas, VA, USA; catalog number TIB-202) was grown in Roswell Park Memorial Institute 1640 Medium-Glutamax-I (RPMI). DMEM and RPMI were supplemented with 0.1% and 10% heat-inactivated fetal calf serum and with an antibiotic/antimycotic solution (10 U/mL penicillin, 10 µg/mL streptomycin, 0.25 μg/mL amphotericin; Thermo Fisher Scientific, Life Technologies SAS, Courtaboeuf, France) and were placed at 37 °C in a humidified atmosphere containing 5% CO_2_. Primary normal human epidermal keratinocytes (NHEK, catalog #00192627) were grown in the KGM^TM^-Gold Bullet Kit, in accordance with the manufacturer’s instructions (Lonza, Basel, Switzerland). Immortalized cell lines were routinely tested to check for an absence of *Mycoplasma* infection. Cells cultured in six- or 96-well polystyrene plates were subjected to pretreatment with peptides or antibodies for 24 h in the dark, at 37 °C. The medium was replaced and the cells were stimulated with a *C. acnes* suspension (MOI = 15), or with the recombinant proteins (nmrCAMP1 or mrCAMP1) for 18 h at 37 °C, under an atmosphere containing 5% CO_2_.

### 4.7. Western Blot

After peptide pretreatment (62.5 μM) for 24 h, followed by stimulation with a *C. acnes* suspension (OD_620 nm_ = 1.0) or the recombinant proteins at a concentration of 10 μg/mL for 18 h, human skin explants were washed in cold sterile PBS. They were shredded for 2 × 1 min with a gentleMACS^TM^ Dissociator in a gentleMACS^TM^ M tube (Miltenyi Biotec, Bergisch Gladbach, Germany) containing RIPA lysis buffer consisting of 50 mM Tris, 150 mM NaCl, 1% NP-40, 0.5% sodium deoxycholate, 0.1% SDS (pH 8.0), 2 mM EDTA, 2 mM sodium pyrophosphate, 10% glycerol, 1 mM sodium orthovanadate, 1 mM phenylmethylsulfonyl fluoride and 10 mg/mL leupeptin. Total protein extract was centrifuged at 10,000× *g* for 15 min at 4 °C and protein concentration was determined by the Lowry method, with BSA as the standard, as described by Peterson [67]. Proteins (30 μg per lane) were fractionated by electrophoresis under denaturing conditions in a NuPAGE Novex 4–12% Bis-Tris gel (1 mm, 10 wells, Invitrogen, UK) and the resulting bands were transferred to nitrocellulose membranes. Membranes were saturated by incubation for 1 h at room temperature in 5% nonfat dry milk in TBS/T buffer (1X TBS, 0.1% Tween-20), and were then incubated for 18 h at 4 °C with rabbit polyclonal IgG antibodies against p-p38 (Catalog #sc-17852-R, Thr 180/Tyr 182, 1:500 BSA), p38 (Catalog #sc-535, C-20), or ERK1 (Catalog #sc94, K-23), with mouse monoclonal IgG antibodies against native and phosphorylated forms of p-ERK (Catalog #sc-7383, E-4), and β-actin (Catalog #sc-47778, C4) used as protein loading controls after dilution to 1:500 in 5% BSA in TBS/T buffer (all antibodies were purchased from Santa Cruz Biotechnology, Inc., Santa Cruz, CA, USA otherwise stated). The membranes were washed and bound antibodies were detected by incubation for 1 h at room temperature with polyclonal goat anti-mouse IgG-horse radish peroxidase (HRP) (Catalog #sc-2005) or goat anti-rabbit IgG-HRP (Catalog #sc-2004), antibodies diluted to 1:2000 in saturation buffer. Unbound material was removed by washing, and peroxidase activity was detected by chemiluminescence with ECL Western Blotting Reagent (Advansta Corp., Menio Park, CA, USA).

### 4.8. Bacterial DNA Extraction

Bacterial DNA was extracted with the E.Z.N.A. Bacterial DNA kit, in accordance with the manufacturer’s recommendations (Omega Bio-Tek, Norcross, GA, USA). The samples were treated with lysozyme to digest the cell wall, and then with protease and RNase. The binding conditions were adjusted and the samples were transferred to spin columns, specifically binding DNA. The columns were washed, the DNA was eluted and its concentration was determined by measuring absorbance at 260 nm (Nanodrop 2000, Thermo Fisher Scientific, Life Technologies SAS, Courtaboeuf, France). The values obtained were in the range of 30 to 220 ng/μL, and the concentration of the solution was adjusted as required.

### 4.9. Production and Purification of the His-Tag Fusion CAMP1 Proteins

We used a phylotype IA1 *C. acnes* strain (75150) expressing the non-mutated CAMP1 protein (nmCAMP1) (GenBank Accession number KX581400), and a phylotype II *C. acnes* strain (27647) expressing the mutated CAMP1 protein (mCAMP1) (GenBank Accession number KX581395) [24]. The gene encoding the CAMP1 factor protein was, in each case, amplified with specific CAMP1F1 (forward and reverse) primers (Appendix A) and the *Pfx* DNA polymerase in a thermocycler ProFlex PCR System (Thermo Fisher Scientific, Life Technologies SAS, Courtaboeuf, France), as follows: initial denaturation at 98 °C for 30 s, 35 cycles of denaturation at 98 °C for 10 s, annealing at 65 °C for 10 s and extension at 72 °C for 30 s. A final extension cycle was performed at 72 °C for 10 min. The CACC sequence was added at the start of the forward primer, to direct the insertion of blunt-ended PCR products into the TOPO cloning vector in accordance with the manufacturer’s instructions (Thermo Fisher Scientific, Life Technologies SAS, Courtaboeuf, France). Both amplicons were purified and inserted into the pET102D/TOPO expression vector (Thermo Fisher Scientific, Life Technologies SAS, Courtaboeuf, France), according to the manufacturer’s instructions, to obtain the vectors pET102-nmCAMP1 and pET102-mCAMP1. For amplification and storage, both recombinant plasmids were introduced into the chemically competent *E. coli* TOP10 strain and selected on the basis of their resistance to ampicillin added to the LB medium (50 μg/mL). Both plasmids were sequenced on a 3730xl DNA Analyzer (Thermo Fisher Scientific, Life Technologies SAS, Courtaboeuf, France) to confirm their sequences. Both recombinant plasmids had the correct sequence and were, therefore, introduced into the BL21 Star^TM^ (DE3) *E. coli* strain for overexpression of the recombinant proteins, nmrCAMP1 and mrCAMP1, with a poly-histidine (6-His) tag at their N-termini. Transformed bacteria were cultured overnight in 1 L of LB medium supplemented with 1% glucose and 50 μg/mL ampicillin. The overnight culture was then used to inoculate 6 L of Overnight Express^TM^ Instant LB medium (Merck, Darmstadt, Germany), which was incubated at 37 °C for four hours for protein production. Bacteria were harvested by centrifugation at 5000× *g* for 10 min and the pellet was resuspended in Lysis Equilibration Buffer (50 mM NaH_2_PO_4_, 300 mM NaCl, pH 8.0 and 1 mg/mL lysozyme). The suspension was subjected to 30-s pulses (10 × 30 bursts with a 30 s period between bursts) of sonication on ice. The lysate was centrifuged for 30 min at 10,000× *g* and the pellet was suspended in a denaturing solubilization buffer (DNS) containing 50 mM NaH_2_PO_4_, 300 mM NaCl and 8 M urea, pH 8.0. The suspension was subjected to sonication (10 pulses of 30 s each), and the resulting lysate was clarified by centrifugation for 30 min at 10,000× *g*. The clarified lysate, containing recombinant proteins, was loaded onto a Protino NI-TED resin immobilized metal chelate affinity chromatography (IMAC) column (GE Healthcare, Chicago, IL, USA). The column was washed with the DNS buffer, and the proteins were eluted in a buffer containing 50 mM NaH_2_PO_4_, 300 mM NaCl, 8 M urea, 250 mM imidazole pH 8.0. The fractions containing the recombinant proteins were analyzed by SDS-PAGE and stored at −20 °C. Before use, recombinant proteins were thoroughly dialyzed against PBS to remove urea and imidazole (Appendix A).

### 4.10. Peptide Array Analysis

For identification of the peptide sequences involved in binding to TLR-2, the amino acid sequences of nmCAMP1 (GenBank Accession number AAS92206.1) and mCAMP1 (GenBank Accession number KX581395) were split into 15-amino acid peptides, with overlapping frames of 11 amino acids, to cover the entire protein sequence. Each peptide was immobilized on a glass slide containing a total of 116 spots (A1 to E20). We also included a positive control corresponding to an unrelated biotinylated peptide (F22) and two negative controls corresponding to unrelated peptides: a V5 epitope tag (F23) and a FLAG tag (F24) (Appendix A) (Active Motif, Carlsbad, CA, USA). The glass slide was incubated in a saturation buffer containing 4 M NaCl, 5% nonfat milk, 0.1% Tween 20, 200 mM Tris (pH 7.6) in TBS for 1 h. The slide was washed three times, for 15 min each, in TBS/T buffer (1 X TBS, 0.1% Tween-20), and incubated overnight at 4 °C with human recombinant TLR-2 (R&D System, Abingdon, UK) diluted at 0.1 μg/mL in TBS/T, with gentle shaking. Unbound antibodies were removed by washing, as described above, and slides were incubated with biotinylated human antibodies against TLR-2 (R&D Systems, Abingdon, UK) diluted to 0.05 μg/mL in TBS/T supplemented with 5% BSA, for 20 h at 4 °C, with gentle mixing. After washing to remove unbound antibodies, the surface protein-TLR-2-biotinylated antibody complexes were detected by incubation for 1 h at room temperature with HRP diluted to 0.5 μg/mL in saturation buffer. Unbound material was removed by washing and peroxidase activity was detected in a chemiluminescence assay (WesternBright ECL, Advansta, Menlo Park, CA, USA).

### 4.11. In Silico CAMP1-3D Structure Modeling

The comparative modeling approach was used to obtain an accurate 3D structure for the nmCAMP1 and mCAMP1 proteins, with the 3D structures of these proteins predicted based on sequence alignments with known protein models in the Protein Data Bank (PDB) database. Reasonable predictions of the structures of these proteins were generated with trRosetta (https://yanglab.nankai.edu.cn/trRosetta/ accessed on 19 March 2021) [68]. All the figures were produced with the molecular visualization system PyMOL [69], Available from: http://www.pymol.org/pymol accessed on 14 January 2021).

### 4.12. Cell Viability Assays

Cell viability was estimated in the (3-[4,5-dimethylthiazol-2-yl]-2,5 diphenyl tetrazolium bromide (MTT) assay, in which cells were incubated with a 0.2% MTT solution in a cell culture medium for 4 h at 37 °C. This solution was removed and dimethyl sulfoxide (DMSO) was added to solubilize the MTT-formazan compound produced in living cells. Absorbance was measured at 550 nm.

### 4.13. Ex Vivo C. acnes-Induced Inflammation Assay

Full-thickness human skin was obtained as excess tissue from abdominoplasty procedures at the De La Tour Clinique in Paris, France. The tissue was collected in an anonymous manner, in accordance with routine procedures approved by the local ethics committee. The subcutaneous fat was trimmed off the skin tissue and punch biopsies (8 mm) were performed on the skin. The discs obtained were immediately placed in DMEM/RPMI/SVF (50/50/20) culture medium containing 2% penicillin/streptomycin and 1% fungicide, and complemented with 250 μg/mL insulin, 0.1 mg/mL epidermal growth factor (EGF) and 0.1 mg/mL fibroblast growth factor (FGF). They were incubated at 37 °C under an atmosphere containing 5% CO_2_ for 18 h, to verify for an absence of bacterial contamination. Skin explants were then incubated with 62.5 μM peptide for 24 h at 37 °C and were then stimulated with a *C. acnes* suspension (OD_620 nm_ = 1.0) or the nmrCAMP1 protein (10 μg/mL) for 18 h at 37 °C. A baseline control was established with untreated and unstimulated skin explants. The positive control was a skin explant stimulated with *C. acnes* suspension or with nmrCAMP1 protein only.

### 4.14. Enzyme-Linked Immunosorbent Assay (ELISA)

Human CXCL8/IL-8 and TNF-α protein concentrations were measured in the supernatants of stimulated cells or skin explants with Ready-Set-Go ELISA Sets (Thermo Fisher Scientific, Life Technologies SAS, Courtaboeuf, France) according to the manufacturer’s instructions. We used serial dilutions of recombinant human CXCL8/IL-8 and TNF-α for the establishment of standard curves. Optical density at 450 nm was determined with a wavelength correction of 570 nm.

### 4.15. Luminex Assay

ProcartaPlex human 9-plex assays (Thermo Fisher Scientific, Life Technologies SAS, Courtaboeuf, France) were used to determine cytokine levels (GM-CSF, CXCL8/IL8, IL-10, IL-17A, IL-1α, IL-1β, IL-12, MMP-1, MMP-3, TNF-α) in the supernatant of NHEK and human skin explants stimulated with *C. acnes* and nmrCAMP1 protein, according to the manufacturer’s specifications.

### 4.16. RNA Isolation and Quantitative Real-Time PCR

Cells and skin explants were grown in 6- or 96-well plates. They were incubated for 24 h with peptides (62.5 µM) or antibodies (1 μg/mL) and were then stimulated by incubation for 18 h with *C. acnes* or nmrCAMP1 (10 μg/mL), as previously described. Total RNA was isolated with TRIzol (Invitrogen, Carlsbad, USA) and the RNeasy^®^ Mini kit and was treated with DNAse I, according to the manufacturer’s instructions (Qiagen, Hilden, Germany). RNA concentration was determined by measuring absorbance at 260 nm on a Nanodrop spectrometer (Labtech, France) and the A_260_/A_280_ ratios for all samples were between 1.6 and 1.9. Complementary DNA was generated from 50 ng of total RNA by reverse transcription at 50 °C for 10 min, and was then subjected to quantitative PCR analysis in a QuantStudio™ 5 Real-Time PCR System thermocycler (Thermo Fisher Scientific, Life Technologies SAS, Courtaboeuf, France) with the iTaq Universal SYBR Green One-Step kit (Bio-Rad Laboratories, Hercules, CA, USA), with two-step cycle conditions as follows: 95 °C for 60 s followed by 40 cycles of 95 °C for 15 s, 68 °C for 60 s, and a final melting curve at 65–95 °C, for 60 s with increments of 0.1 °C/s. The threshold cycle (Ct) values for the genes studied were determined from the amplification curves. The amount of RNA in stimulated cells relative to control cells was calculated by the 2ΔCt method and is expressed as a relative fold change in expression normalized against the expression of an internal control gene (*GAPDH*). All the primers used in this study are described in Appendix A.

### 4.17. RNA Sequencing Analysis

Human skin explants were left untreated and unstimulated (negative control group), were stimulated with nmrCAMP1 (10 μg/mL) for 24 h at 37 °C (positive control group) or were treated with the CAMP1-related B2 peptide (62.5 μM) for 24 h at 37 °C and then stimulated with nmrCAMP1 (treated group). Total RNA was extracted with TRIzol (Invitrogen, Carlsbad, USA) and the RNeasy^®^ Mini kit and was treated with DNAse I, according to the manufacturer’s instructions (Qiagen, Hilden, Germany). We used 250 ng of RNA to generate the cDNA library with the Illumina Truseq mRNA prep ligation kit. RNAseq was performed on an Illumina NovaSeq instrument providing 50 bp paired-end reads. The reads (about 26 to 34 million reads per sample) in the fastq files were aligned with the GRCh38.101 version of the human genome with RSEM (v1.3.3), using STAR aligner (v2.7.6a). Differential expression between the two sample groups was analyzed with the DESeq2 package (https://github.com/BSGenomique/genomic-rnaseq-pipeline/releases/tag/v1.0420 accessed on 22 October 2021) with an adjusted *p*-value threshold of 0.05. RNA sequencing results were analyzed with the GENOM’IC’s application (https://github.com/GENOM-IC-Cochin/shiny-rnaseq-viz accessed on 26 November 2021). (GENOM’IC platform, Institut Cochin, U1016, Paris, France). Kyoto Encyclopedia of Genes and Genomes (KEGG) pathway enrichment analyses were performed for the upregulated and downregulated genes with the Database for Annotation, Visualization, and Integrated Discovery (DAVID) (https://david.ncifcrf.gov/ accessed on 21 March 2022). For IL-6, CXCL1, CXCL2, CXCL3, CCL3 and CCL4, differential expression was confirmed by qRT-PCR with TaqMan probes labeled with FAM fluorochromes. Amplifications were performed in the presence of 50 ng cDNA (previously reverse transcribed with the PreSuperScript™ IV VILO™ Master Mix and ezDNase™ Enzyme), 10 μL TaqMan™ Fast Advanced Master Mix, 1 μL TaqMan™ assay 20X (ACTB: Hs99999903_m1, IL-6: Hs00985639_m1, CXCL1: Hs00236937_m1, CXCL2: Hs00236966_m1, CXCL3: Hs00171061_m1, CCL3: Hs00234142_m1 or CCL4: Hs99999148_m1), in a final volume of 20 μL (Thermo Fisher Scientific, Life Technologies SAS, Courtaboeuf, France). The reaction was performed on a QuantStudio™ 5 Real-Time PCR System thermocycler (Thermo Fisher Scientific, Life Technologies SAS, Courtaboeuf, France) with an initial UNG incubation for 2 min at 50 °C, one polymerase activation step at 95 °C for 20 s, followed by 40 cycles of denaturation for 1 s at 95 °C, annealing and extension for 20 s at 60 °C. The same method as described above was used to determine the fold-change in expression relative to the internal control ACTB.

### 4.18. Statistical Analysis

The significance of differences between experimental groups was determined in parametric unpaired *t*-tests and ANOVA, implemented in GraphPad Prism 9 (GraphPad, La Jolla, CA, USA). A *p*-value ≤ 0.05 was considered statistically significant, and significance is indicated as follows: * (*p* ≤ 0.05), ** (*p* ≤ 0.01), *** (*p* ≤ 0.001) and **** (*p* ≤ 0.0001).

## 5. Conclusions

In summary, we characterized CAMP factor 1 as a proinflammatory protein produced by *C. acnes*. Specific antibodies against CAMP1 decreased *C. acnes*-induced CXCL8/IL-8 production by keratinocytes in vitro. We confirmed the specific recognition of CAMP1 by TLR-2, with recombinant CAMP1 protein, and we identified five CAMP1-related peptides involved in this binding. We also showed that these CAMP1-related peptides were anti-inflammatory, as they inhibited the production of proinflammatory cytokines and chemokines in vitro and ex vivo in human skin explants by inhibiting the p38 MAPK. We selected one of these peptides for further analysis and performed an RNA sequencing analysis on a treated skin explant. All the proinflammatory molecules induced by rCAMP1 stimulation were downregulated in the presence of the selected peptide. This CAMP1-related peptide also appeared to be harmless for skin-related bacteria, as it did not affect their growth. *C. acnes* is able to trigger inflammation reaction through the TLR-2 pathway and TLR-2 is highly expressed in acne lesions; therefore, this candidate peptide appears to be a new TLR-2 modulator that could be clinically tested on patients presenting inflammatory acne lesions.

## Figures and Tables

**Figure 1 ijms-23-05065-f001:**
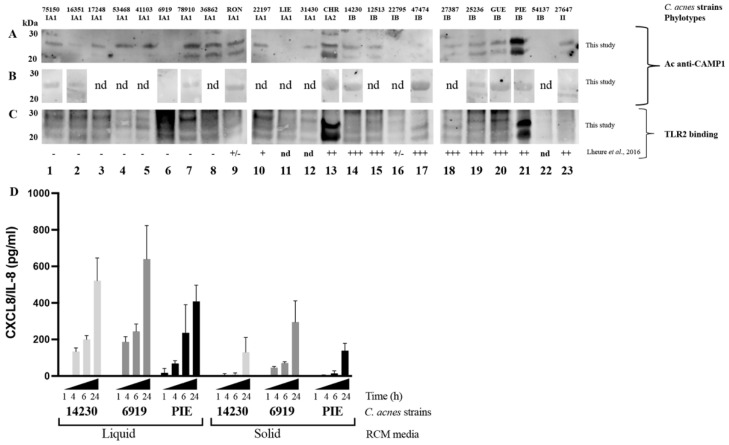
Expression of Christie–Atkins–Munch–Petersen (CAMP)1 in *Cutibacterium acnes* (*C. acnes*) strains. *C. acnes* surface proteins were extracted from a five-day culture (**A**,**C**) in liquid RCM medium or (**B**) on solid RCM medium and separated by electrophoresis in a 4–12% NuPAGE SDS BisTris gel (50 μg). Separated proteins were transferred onto nitrocellulose membranes, which were incubated (**A**,**B**) with antibodies against CAMP1 or (**C**) with recombinant toll like receptor (TLR)-2. TLR-2 binding activity was detected with specific biotinylated antibodies against TLR-2, as described in the Materials and Methods. TLR-2 binding activity previously reported by Lheure et al. [24]. Lanes 1 to 12 correspond to phylotype IA1 strains: 75150, 16351, 17248, 53468, 41103, 6919, 78910, 36862, RON, 22197, LIE and 31430, respectively. Lane 13 corresponds to phylotype IA2 strain: CHR. Lanes 14 to 22 correspond to phylotype IB strains: 14230, 12513, 22795, 47474, 27387, 25236, GUE, PIE and 54137, respectively. Lane 23 corresponds to phylotype II strain: 27647. (**D**) HaCaT cells were stimulated for 1, 4, 6 and 24 h at 37 °C, with *C. acnes* strains 6919 (phylotype IA1), 14230 (phylotype IB) or PIE (phylotype IB) grown in liquid or on solid medium. C-X-C motif chemokine ligand interleukin (CXCL)8/(IL)-8 protein levels were assessed by enzyme-like immunosorbent assay (ELISA). nd: non-determined.

**Figure 2 ijms-23-05065-f002:**
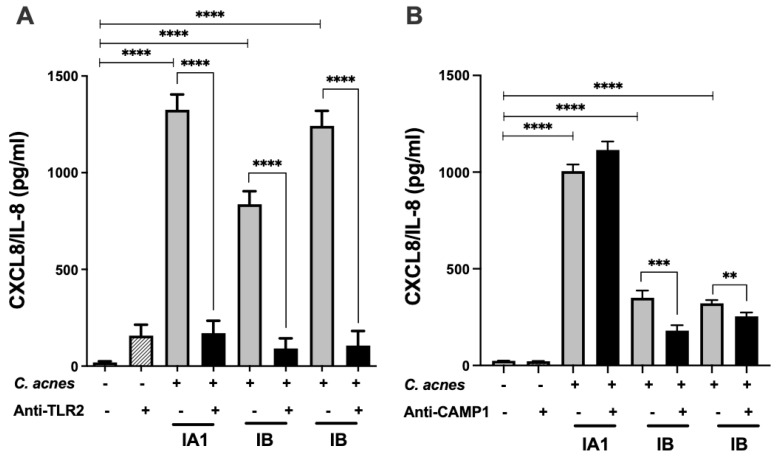
Inhibition of *C. acnes*-induced CXCL8/IL-8 production by TLR-2 and CAMP1 antibodies. HaCaT cells were treated with (**A**) anti-TLR-2 antibody (1 μg/mL) or (**B**) anti-CAMP1 antibody (1 μg/mL) for 24 h (gray bar), and were then stimulated with various strains of *C. acnes* (multiplicy of infection 15 (MOI 15) (phylotype IA1, strain 6919; phylotype IB, strains PIE and 14230) for 18 h (dark bar). Control experiments correspond to *C. acnes* stimulation of HaCaT cells only (gray bar). CXCL8/IL-8 production was measured by ELISA. Statistical significance is indicated by ** (*p* < 0.01), *** (*p* < 0.001) and **** (*p* < 0.0001).

**Figure 3 ijms-23-05065-f003:**
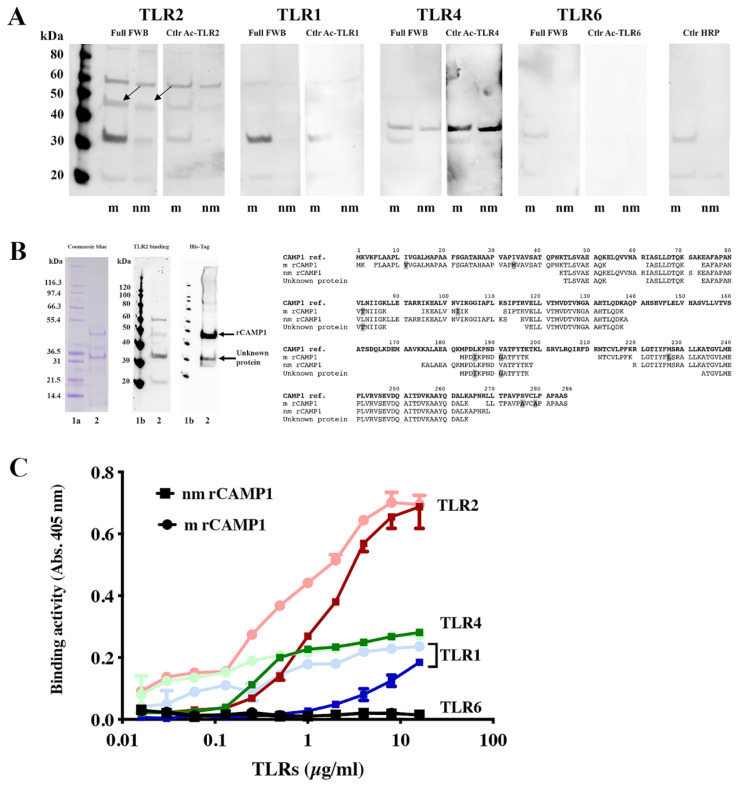
Specific recognition of *C. acnes* recombinant CAMP1 by TLR-2. (**A**) The recombinant proteins nmrCAMP1 (nm) and mrCAMP1 (m) (50 μg) were separated by electrophoresis in a 4–12% NuPAGE SDS BisTris gel and transferred onto nitrocellulose membranes, which were incubated with recombinant TLR-2, TLR-1, TLR-4 and TLR-6 (0.1 μg/mL). TLR binding activity was detected with specific biotinylated antibodies against TLR-2, TLR-1, TLR-4 and TLR-6, respectively, as described in the Materials and Methods. The arrow indicates the position of the 47 kDa band of interest. (**B**) The recombinant proteins were separated by electrophoresis in a 4–12% NuPAGE SDS BisTris gel (50 μg) with detection by Coomassie brilliant blue staining. The separated proteins were transferred onto nitrocellulose membranes, which were incubated with poly-histidine antibody or recombinant TLR-2 (0.1 μg/mL), with the detection of TLR binding activity with specific biotinylated antibodies against TLR-2. Lanes 1a and 1b contain the molecular mass markers. Arrows indicate the positions of the 30 and 47 kDa bands of interest. The peptide sequences of the 47 kDa nmrCAMP1 and mrCAMP1 and the 30 kDa protein were obtained by LC-MS/MS (in bold, the amino acid substitutions in mrCAMP1). (**C**) Immobilized nmrCAMP1 (square) and mrCAMP1 (circle) (20 μg/mL) were probed with various concentrations of TLR-1, TLR-2, TLR-4 or TLR-6 (0.016 to 16 μg/mL).

**Figure 4 ijms-23-05065-f004:**
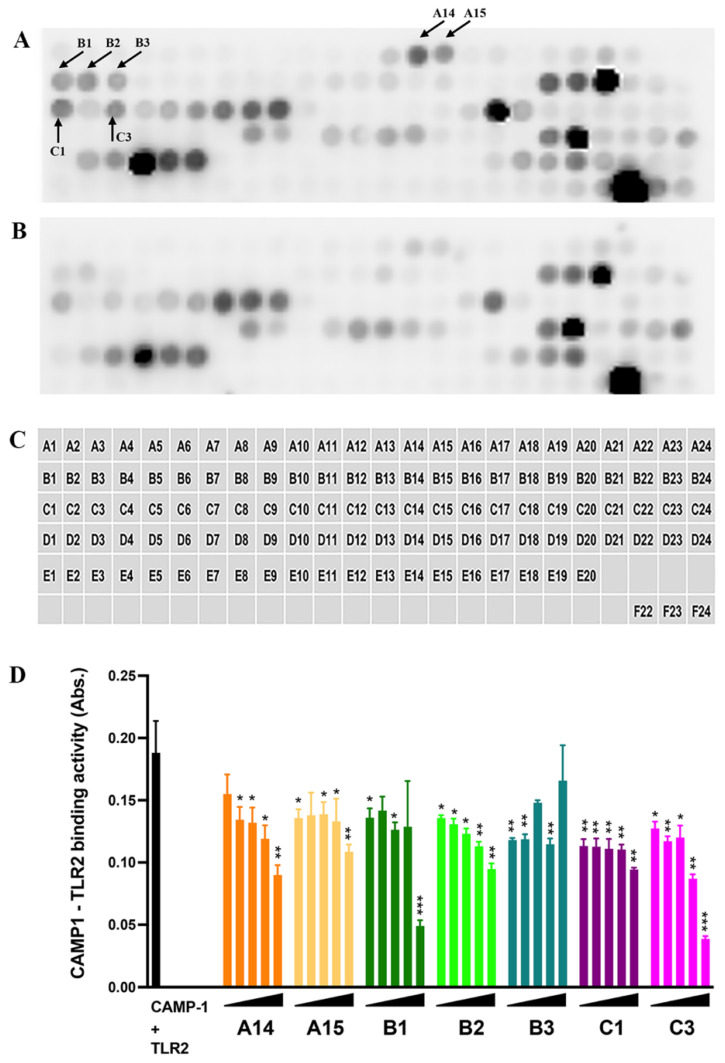
Subpeptides from CAMP1 recognized by TLR-2. (**A**,**B**,**C**) The peptides derived from CAMP1 proteins: group F (positions A1-C21), group A (positions C22-E9), group C (positions E10-E13), group D (positions E14-E17), group E (positions E18-E20); a biotinylated peptide used as a positive control (position F22) and a peptide used as a negative control (position F24), were immobilized on glass plates. (**A**) The plates were incubated with TLR-2 (10 μg/mL)/anti-TLR-2 antibody (0.1 μg/mL)/HRP (0.5 μg/mL) and (**B**) with anti-TLR-2 antibody (0.1 μg/mL)/HRP (0.5 µg/mL). The arrows indicate the positions of peptides specifically recognized by TLR-2. (**D**) Immobilized rCAMP1 (20 μg/mL) proteins were probed with biotinylated TRL-2 (0.5 μg/mL) (dark bar) after pre-treatment with peptide (A14, A15, B1, B2, C1 and C3) at various concentrations (1.25, 12.5, 125, 1250 and 12,500 nM). Statistical significance is indicated by * (*p* < 0.05), ** (*p* < 0.01) and *** (*p* < 0.001).

**Figure 5 ijms-23-05065-f005:**
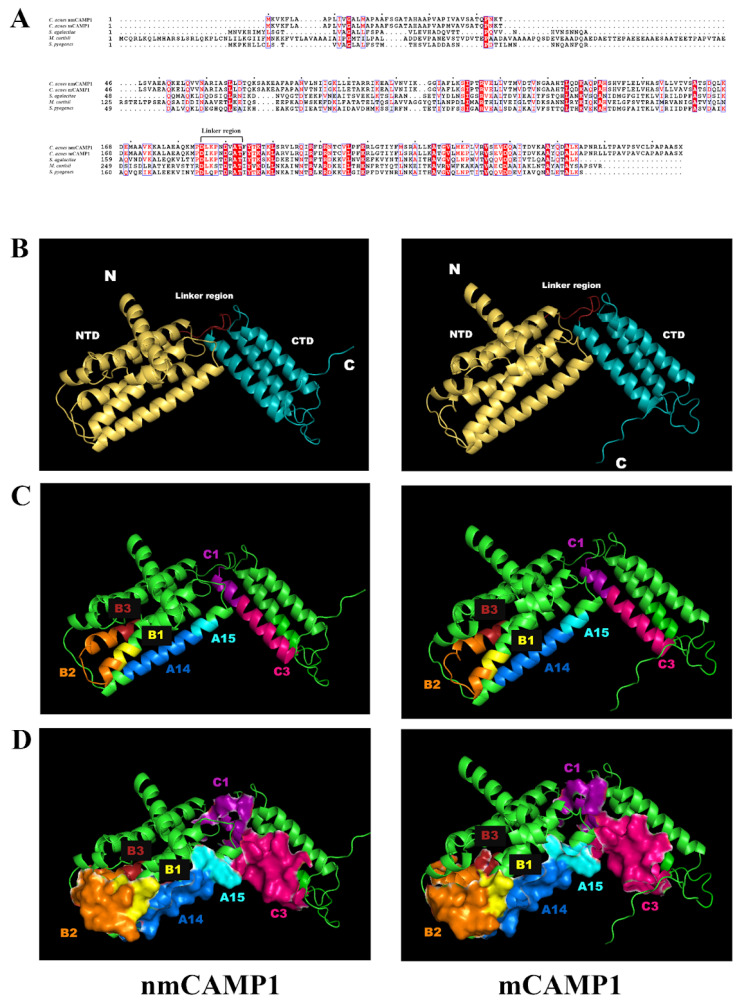
*In silico* 3D analysis of *C. acnes* factor CAMP1. (**A**) Sequence alignment of CAMP factors from different species. Conservatively substituted residues are boxed and strictly conserved residues are highlighted with a red background. The figure was constructed with ESPript 3.0. Reference sequences: *Cutibacterium acnes* nmCAMP1: AAS92206.1; *Cutibacterium acnes* mCAMP1: KX581410; *Streptococcus agalactiae*: ZP_08649639.1; *Mobiluncus curtisii*: YP_003718285.1; *Streptococcus pyogenes*: WP_111681137.1. (**B**–**D**) Predicted structures of the nmCAMP1 and mCAMP1 proteins obtained by homology modeling with the trRosetta server and the images generated with PyMOL software. (**B**) Cartoon representation of CAMP1. The NTD domain is colored in yellow, the CTD domain in cyan and the linker region in red. (**C**) CAMP1 peptides involved in TLR-2 binding are highlighted: A14 in blue, A15 in cyan, B1 in yellow, B2 in orange, B3 in red, C1 in purple and C3 in pink. (**D**) Surface representation of CAMP1 peptides.

**Figure 6 ijms-23-05065-f006:**
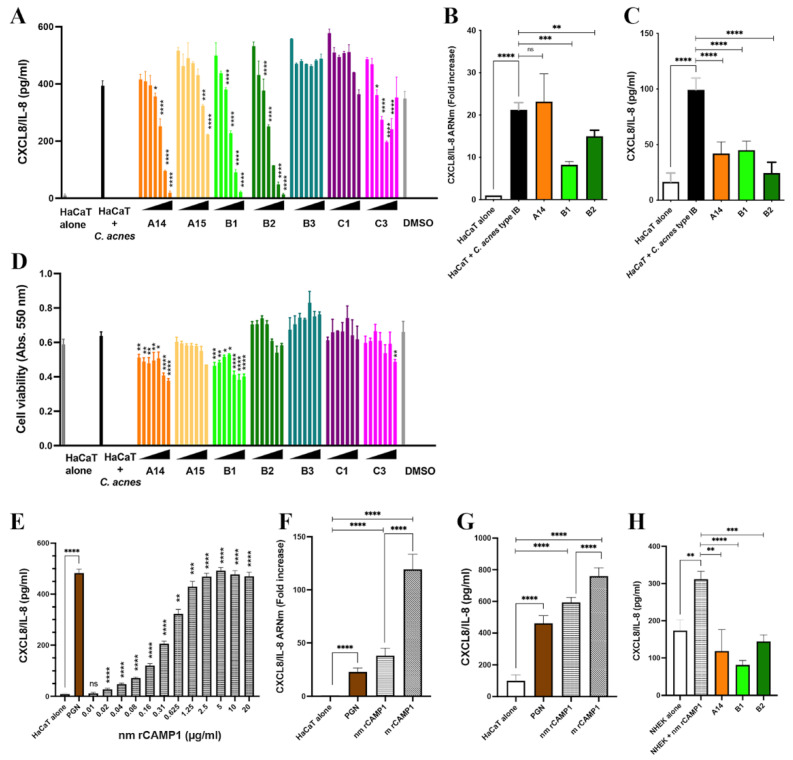
Inhibition of CXCL8/IL-8 production by CAMP1-derived peptides in vitro. (**A**–**C**) HaCaT cells were treated with A14, A15, B1, B2, B3, C1 and C3 peptides at a concentration of 3.9 to 250 μM or with A14, B1, B2 peptides (62.5 μM) for 24 h and were then stimulated with *C. acnes* (MOI 15) for 18 h. The negative control consists of untreated and unstimulated cells (HaCaT alone). The positive control corresponds to cells stimulated with *C. acnes* (HaCaT + *C. acnes*). (**D**) Cytotoxicity was assessed in the MTT assay. (**A**,**D**) Dimethy sulfoxide (DMSO) corresponds to the control experiment with a solution of DMSO/H_2_O 30:70 (v/v) used to solubilize the peptides A14 and A15. (**E**) HaCaT cells were stimulated with various concentrations of nmrCAMP1 (0.01 to 20 μg/mL). The positive control corresponds to cells stimulated with PGN (5 μg/mL). (**F**,**G**) HaCaT cells were stimulated with nmrCAMP1 and mrCAMP1 (10 μg/mL). The positive control corresponds to cells stimulated with PGN (5 μg/mL). (**H**) Primary normal human epidermal keratinocyte cells (NHEK) were treated with A14, B1 and B2 peptides at a concentration of 62.5 μM for 24 h at 37 °C and stimulated with nmrCAMP1 (10 μg/mL) for 18 h. The negative control was the untreated and unstimulated cells (NHEK alone). The positive control corresponds to cells stimulated with nmrCAMP1 (NHEK + nmrCAMP1). (**B**,**F**) The level of CXCL8/IL-8 mRNA was assessed by qRT-PCR. (**A**,**C**,**E**,**G**,**H**) CXCL8/IL-8 production was assessed by ELISA. Statistical significance is indicated by * (*p* < 0.05), ** (*p* < 0.01), *** (*p* < 0.001) and **** (*p* < 0.0001).

**Figure 7 ijms-23-05065-f007:**
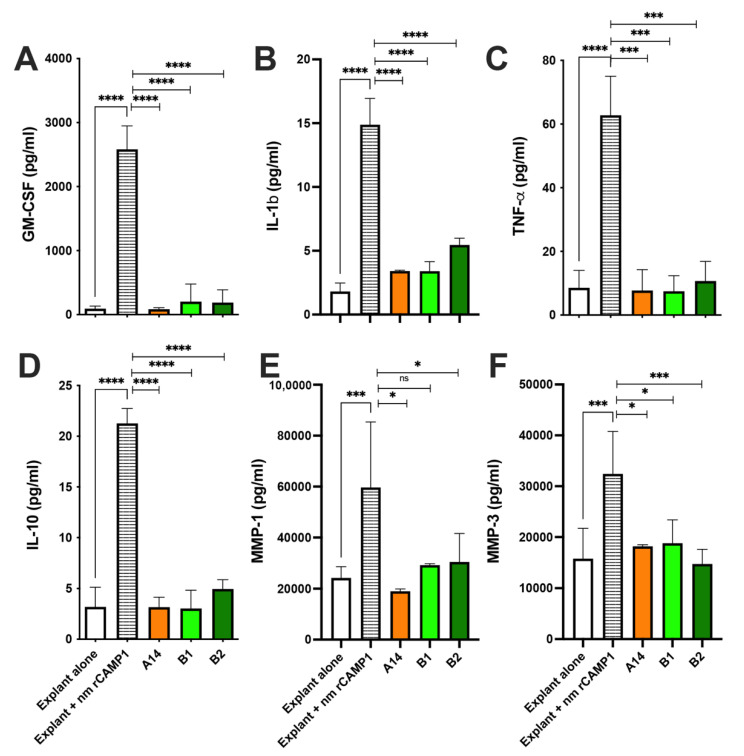
CAMP1-related peptides inhibit the production of nmrCAMP1-induced proinflammatory molecules ex vivo. Human skin explants were left untreated and unstimulated (Explant alone), were stimulated with nmrCAMP1 (10 μg/mL) (Explant + nmrCAMP1), and treated with A14, B1 and B2 peptides (62.5 μM) for 24 h and then stimulated with nmrCAMP1 for 18 h. (**A**–**F**) GM-CSF, IL-1β, TNF-α, IL-10, MMP-1 and MMP-3 levels were measured by ELISA in the culture supernatant. The data shown are means ± SEM (*n* = 6). Statistical significance is indicated by * (*p* < 0.05), *** (*p* < 0.001), and **** (*p* < 0.0001).

**Figure 8 ijms-23-05065-f008:**
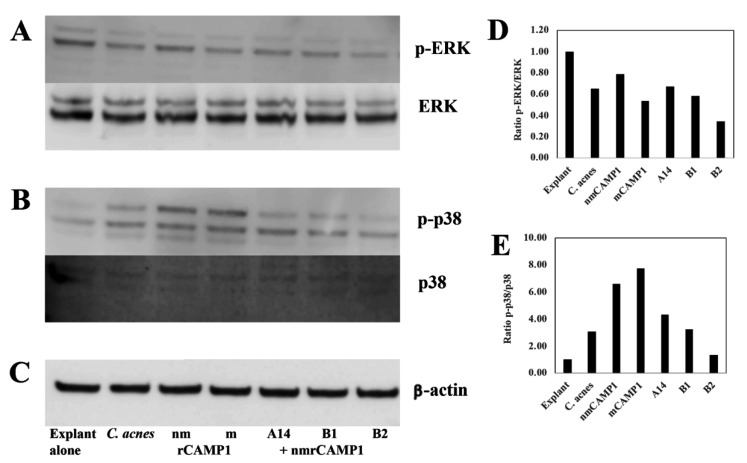
CAMP1-derived peptides inhibit the MAPK pathway. Human skin explants were treated for 24 h with A14, B1 or B2 peptide (62.5 μM) and were then stimulated with *C. acnes* (OD_620 nm_ = 1.0) for 18 h or with nmrCAMP1 or mrCAMP1 (10 μg/mL). Human skin explant lysates were analyzed by Western blotting with specific antibodies against (**A**) p-ERK, ERK, (**B**) p-p38, p38 and (**C**) β-actin. Blots loaded with the same protein samples were run at the same time. Densitometry analysis of the ratio of expression levels for (**D**) p-ERK/ERK, (**E**) p-p38/p38.

**Figure 9 ijms-23-05065-f009:**
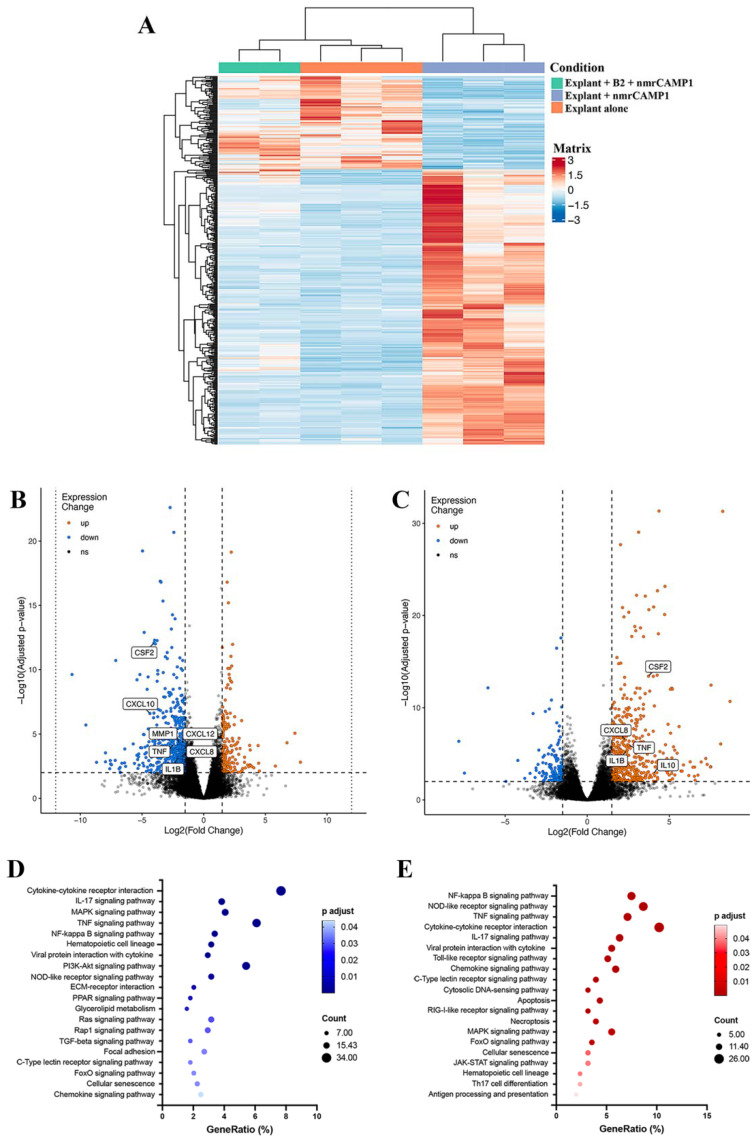
RNA sequencing analysis for nmrCAMP1-induced inflammation and after B2 CAMP1-related peptide treatment. Screening and enrichment analysis on differentially expressed mRNAs in human skin explants treated with B2 CAMP1-related peptide and stimulated with nmrCAMP1 relative to human skin explants stimulated with nmrCAMP1. (**A**) Cluster analysis for mRNAs between the unstimulated and untreated group (*n* = 3), the stimulated group (*n* = 3) and the treated and stimulated group (*n* = 2). (**B**,**C**) Volcano plots on total mRNA between the treated and control groups (thresholds *p* < 0.05 and log2(fold-change) > 1.5 in either direction). (**D**,**E**) KEGG pathway enrichment analysis performed with the David Bioinformatics Resource on the downregulated and upregulated gene groups, respectively (thresholds *p* < 0.05, log2(fold-change) > 1.5 in either direction, and basal state mean > 100).

**Figure 10 ijms-23-05065-f010:**
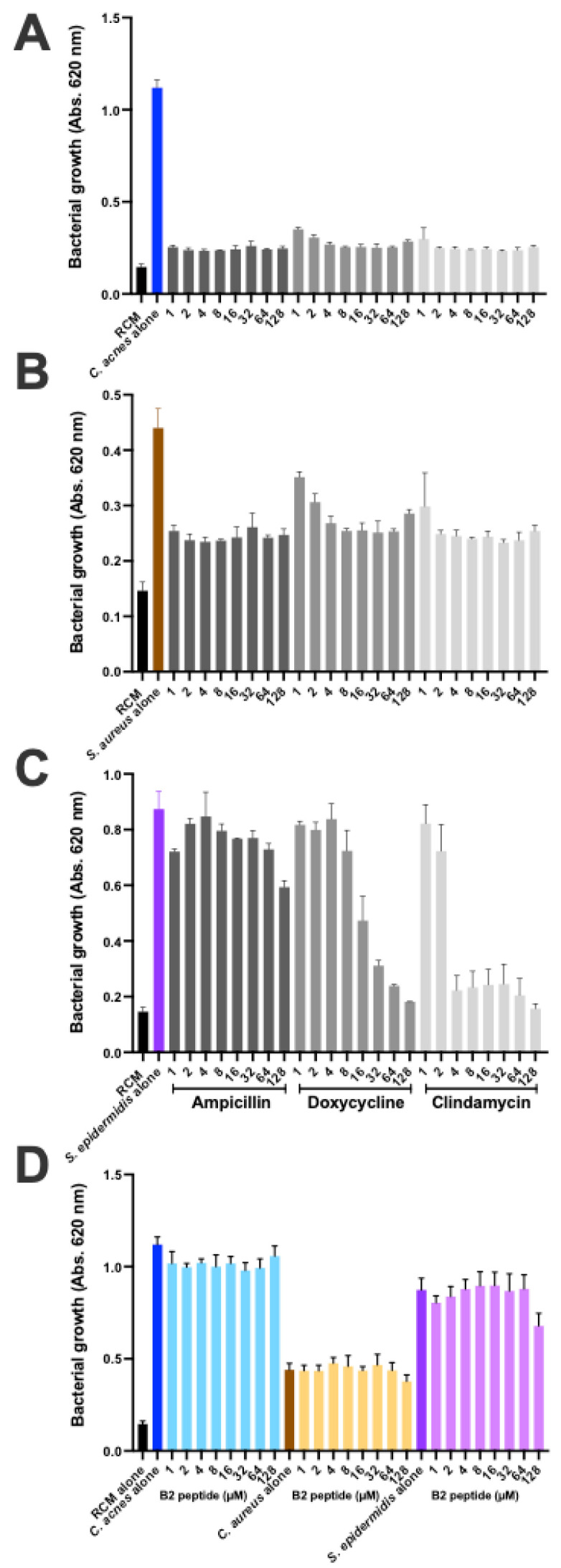
The CAMP1-derived B2 peptide preserves bacterial growth. *C. acnes* 6919, *S. aureus* and *S. epidermidis* were grown in the presence of (**A**–**C**) antibiotic (ampicillin, doxycycline, clindamycin) and (**D**) the B2 peptide, for 5 days and 24 h, respectively. Bacterial growth was monitored at 620 nm.

**Table 1 ijms-23-05065-t001:** CAMP1 peptide sequence recognized by TLR-2.

Position in Plate	Peptide Sequence ^a^	Nature of the Mutation ^b^	CAMP1 Group
A14	K-E-L-Q-V-V-N-A-R-I-A-S-L-L-D	/	A
A15	V-V-N-A-R-I-A-S-L-L-D-T-Q-K-S	/	A
B1	E-A-L-V-N-**I***-I-K-G-G-**V**-A-F-L-K	V102 > I; I107 > V	F
B2	N-**I**-I-K-G-G-**V**-A-F-L-K-S-I-P-T	V102 > I; I107 > V	F
B3	G-G-**V**-A-F-L-K-S-I-P-T-R-V-E-L	I107 > V	F
C1	T-F-Y-T-K-**A**-K-L-**A**-R-V-L-R-Q-I	T198 > A; S201 > A	F
C3	**A**-R-V-L-R-Q-I-R-F-D-R-N-T-C-V	S201 > A	F

^a^ Peptide sequences were compared to the reference sequence of *C. acnes* 6919 (AAS92206.1). ^b^ [24]. * Mutated amino acids (bold) relative to the reference sequence of *C. acnes* 6919 (AAS92206.1).

**Table 2 ijms-23-05065-t002:** The 10 genes displaying the highest levels of differential expression for the 12 common KEGG pathways ^a^.

Gene Name	nmrCAMP1 Stimulation		B2-Peptide-Treated + nmrCAMP1 Stimulation	
	log2(Fold-Change)	P-Adjust	log2(Fold-Change)	P-Adjust
** *IL6* **	2.20	4.33 × 10^−7^	−1.64	1.67 × 10^−3^
** *TNF* **	2.55	1.20 × 10^−6^	−2.42	6.42 × 10^−5^
** *CXCL1* **	1.99	1.09 × 10^−5^	−2.07	6.07 × 10^−5^
** *CXCL2* **	2.94	1.61 × 10^−10^	−2.16	9.15 × 10^−5^
** *CXCL3* **	2.22	1.23 × 10^−7^	−2.23	3.86 × 10^−6^
** *CXCL8* **	2.32	3.06 × 10^−7^	−1.85	5.91 × 10^−4^
** *IL1B* **	2.39	4.99 × 10^−6^	−2.06	7.17 × 10^−4^
** *CCL2* **	2.65	2.36 × 10^−2^	−2.89	2.50 × 10^−2^
** *CCL4* **	4.17	2.60 × 10^−4^	−3.09	2.21 × 10^−2^
** *CSF2* **	3.73	3.93 × 10^−14^	-4.03	8.42 × 10^−13^

^a^ The 12 common KEGG pathways: cytokine–cytokine receptor interaction, IL-17 signaling pathway, TNF signaling pathway, MAPK signaling pathway, NF-kappa B signaling pathway, hematopoietic cell lineage, viral protein interaction with cytokine and cytokine receptor, NOD-like receptor signaling pathway, C-type lectin receptor signaling pathway, FoxO signaling pathway, cellular senescence and chemokine signaling pathway.

## Data Availability

The data contained within the article and the original data supporting the finding of this study are available from the corresponding author upon reasonable request.

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
