# Peer review of "Characterization of a Cutibacterium acnes Camp Factor 1-Related Peptide as a New TLR-2 Modulator in In Vitro and Ex Vivo Models of Inflammation"

_ijms, 2022, doi:10.3390/ijms23095065_

Round 1
Reviewer 1 Report
In the presented manuscript authors identified and described CAMP1 peptide as a TLR-2 modulator. The study includes considerable number of assays. Results are well presented and conclusion is supported by the evidence.
Language is fine - minor text required.
Minor questions: why absorbance of MTT was evaluated at 540 nm?
Author Response
In the presented manuscript authors identified and described CAMP1 peptide as a TLR-2 modulator. The study includes considerable number of assays. Results are well presented and conclusion is supported by the evidence.
Language is fine - minor text required.
Minor questions: why absorbance of MTT was evaluated at 540 nm?
Answer:
- The manuscript has been carefully read
- The MTT was measured at 550 nm, we corrected the value in the Material & Method section in lane 840
Reviewer 2 Report
I really liked this paper and found it scientifically sound. I do not that in the first paragraph, you mention C. acnes is anerobic, but you should add that is aerotolerant as well (as this can be significant clinically).
I also would like you to expand on your conclusion as to how you believe a TLR-2 modulator might be functional in acne therapy.
Author Response
I really liked this paper and found it scientifically sound.
I do not that in the first paragraph, you mention C. acnes is anerobic, but you should add that is aerotolerant as well (as this can be significant clinically).
I also would like you to expand on your conclusion as to how you believe a TLR-2 modulator might be functional in acne therapy.
Answer:
- We agreed that C. acnes is able to growth under normal atmosphere, consequently we added the aerotolerant term in lane 35.
- We added a sentence in the conclusion about the test of this peptide on a acne patients:
“C. acnes is able to trigger inflammation reaction through the TLR-2 pathway and TLR-2 is highly expressed in acne lesion. Therefore, this candidate peptide appears to be a new TLR-2 modulator which could be clinically tested on patients presenting inflammatory acne lesions.”